# Less Forgetting, More OOD Generalization: Adaptive Augmented Reweighted Replay (AA-RR) for Continual Learning

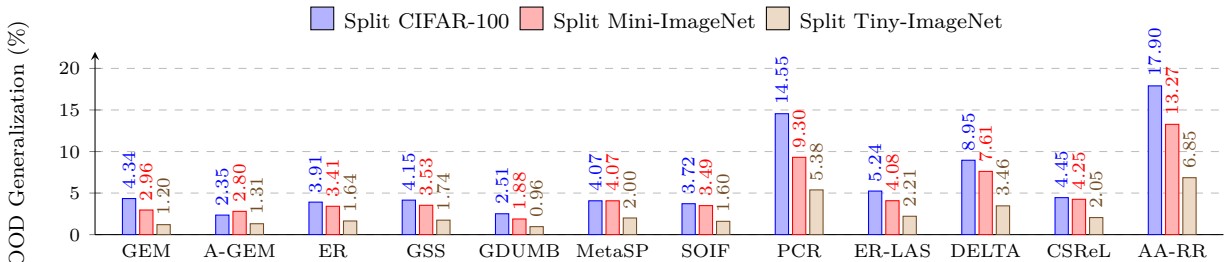

Figure 1: **Evaluating Out-of-Distribution (OOD) Generalization:** State-of-the-art rehearsal-based methods exhibit a significant performance drop ~~on OOD samples~~ **under label-preserving covariate shifts** across Split CIFAR-100, Split Mini-ImageNet, and Split Tiny-ImageNet, revealing limited ~~generalization capability~~ **OOD robustness**. The proposed AA-RR method effectively addresses this limitation, achieving consistent improvements across all benchmarks.

## Abstract

Machine learning models often forget previously learned classes when trained sequentially. Rehearsal-based methods mitigate this by replaying stored samples, but their reliance on memorization leads to poor out-of-distribution (OOD) generalization—a problem that remains largely unstudied. This memorization is driven by unbalanced gradient updates, spurious correlations, and class-imbalanced replay buffers. To address these issues, we introduce Adaptive Augmented Reweighted Replay (AA-RR), a lightweight framework designed to improve ~~generalization~~ **both in-distribution retention and OOD robustness** in rehearsal-based continual learning (CL). AA-RR applies adaptive, class-aware loss reweighting to correct gradient imbalance while accounting for data recency and limited buffer capacity. It further incorporates data-centric augmentation and a principled sample-selection strategy based on forgetting dynamics to retain representative, consistently learned examples. Experiments on standard CL benchmarks show that AA-RR ~~markedly boosts generalization and surpasses state-of-the-art baselines, especially under covariate shift~~ **substantially improves robustness under covariate shift while remaining competitive, and often improving, under standard i.i.d. evaluation.**[1]

## 1 Introduction

Continual Learning (CL) enables models to learn new tasks or classes sequentially without forgetting previously acquired knowledge. Approaches to CL include regularization-based methods, which constrain parameter updates to preserve prior knowledge Kirkpatrick et al. (2017); Chaudhry et al. (2018a); architecture-based methods, which expand or adapt network structures for new tasks Mallya & Lazebnik (2018); Hung et al.

---

[1]Code is available in the supplementary materials.

(2019); and rehearsal-based methods, which mitigate forgetting by replaying stored samples from earlier tasks during training Sun et al. (2022; 2023); Raghavan et al. (2024); Tong et al. (2025).

Among these approaches, rehearsal-based methods have attracted significant attention due to their simplicity and strong effectiveness. However, despite their strong performance, these methods remain underexplored in terms of robustness, especially regarding out-of-distribution (OOD) generalization. Some studies Verwimp et al. (2021); Zhang et al. (2022b) have raised concerns that such methods may rely heavily on memorizing replayed samples rather than truly learning and generalizing the underlying knowledge. Despite growing attention to **standard (i.i.d.)** generalization and overfitting in CL Bonicelli et al. (2022); Yu et al. (2022), the challenge of OOD generalization remains largely unexplored**, particularly for rehearsal-based CL methods trained on clean sequential tasks and evaluated under label-preserving covariate shifts**. ~~Our investigations clearly demonstrate that current approaches fail to generalize to samples exhibiting covariate shifts (see Figure 1). Notably, by design, CL methods aim to retain knowledge from previously learned classes/tasks — in other words, to prevent forgetting. However, the critical point is that preventing forgetting should not come at the cost of memorization, which can severely degrade a model's ability to generalize, particularly to OOD data. In essence, while CL methods should preserve past knowledge, they must do so in a way that maintains the model's generalization capability. This overlooked aspect motivates our study, where we analyze the problem and propose an effective solution.~~ **Our investigations show that current rehearsal-based approaches often suffer a substantial performance drop on corrupted test samples, even when they retain reasonable performance under standard i.i.d. evaluation (see Figure 1). Thus, our focus is not generalization in CL broadly, but the specific question of whether rehearsal-based CL methods preserve knowledge in a way that remains robust beyond the training distribution. This distinction is important because preventing forgetting and improving OOD robustness are related but not identical goals. A model may retain past-task performance under the original test distribution while still relying on brittle or spurious features that fail under covariate shift. Conversely, improving OOD robustness requires not only learning robust representations within each task, but also preserving such representations throughout the CL sequence. This OOD-focused aspect motivates our study, where we analyze how rehearsal dynamics affect retention and robustness, and propose an effective solution.**

We identify three sources of *overfitting* **that affect i.i.d. retention and OOD robustness** in buffer-based CL. First, unbalanced gradient propagation due to sequential training, ~~leading to dominance of recent tasks~~ **causes recent tasks to dominate optimization, which primarily explains forgetting and degradation under standard i.i.d. evaluation but also affects the preservation of robust features learned from earlier tasks**. Second, spurious correlations **can** arise within individual training stages— ~~patterns that fit the current domain but fail to generalize~~**causing representations to fit the current domain while failing under covariate shift**. Third, buffer distribution **bias**, ~~depending on~~ **determined by** which samples are retained and whether class and task balance are ~~maintained~~ **preserved, affects both stable retention and robustness**.

Buffer-based CL inherently suffers from class imbalance, as earlier task examples are increasingly underrepresented due to limited buffer capacity. This skews optimization toward recent classes, exacerbating forgetting. A common strategy to address this issue is to reweight the softmax cross-entropy loss to counteract unbalanced gradient contributions. Recent works Lin et al. (2023); Huang et al. (2023); Raghavan et al. (2024) adopt this approach by reweighting the loss based on class frequency, thereby mitigating the skewed gradient propagation.

However, existing reweighting strategies Lin et al. (2023); Huang et al. (2023) have critical limitations. They rely on batch-wise class frequency, providing no gradient signal for absent classes—a side effect of inheriting long-tail learning techniques designed for stationary settings. Moreover, they Lin et al. (2023); Huang et al. (2023); Raghavan et al. (2024) do not consider the temporal ordering (age or recency) of buffered samples, despite its importance for balancing stability and plasticity in CL.

Sample selection policies in rehearsal methods also play a central role in preventing forgetting. Most approaches use random sampling to update the memory buffer Lopez-Paz & Ranzato (2017); Chaudhry et al. (2018b; 2019); Prabhu et al. (2020); Lin et al. (2023); Huang et al. (2023); Raghavan et al. (2024). However,

without principled selection, such strategies are prone to storing outliers or non-representative examples, which impairs generalization. Random sampling also tends to produce class-imbalanced buffers (Section 5.2, Table 2), leading to long-tailed distributions that distort decision boundaries Cui et al. (2019); Samuel & Chechik (2021); Shi et al. (2023).

To overcome these issues, more principled selection strategies have been proposed, such as those based on gradient informativeness Aljundi et al. (2019); Jin et al. (2021); Yoon et al. (2021); Tiwari et al. (2022), Shapley values Shim et al. (2021), and influence functions Sun et al. (2022; 2023), which aim to prioritize informative samples during buffer updates. Nevertheless, these methods still struggle with class and task imbalance (Section 5.2, Table 2) and often incur significant computational costs due to gradient tracking or auxiliary optimization. Even the more recent gradient-free coreset methods Tong et al. (2025) reduce some overhead but still require auxiliary model training for each task (Section 5.2, Table 3).

To overcome the above limitations, we propose **A**daptive **A**ugmented **R**eweighted **R**eplay (**AA-RR**)—a lightweight yet effective framework that enhances generalization in rehearsal-based CL.

AA-RR introduces a class-aware cross-entropy softmax reweighting mechanism that adaptively adjusts loss contributions to counter class imbalance caused by non-stationary data, recency effects, and constrained buffer size. This reduces gradient imbalance across classes and promotes stable learning over time.

In addition, we extend insights from data-centric generalization strategies in stationary settings to the CL paradigm, aiming to mitigate overfitting caused by spurious correlations. Prior work in stationary training has shown that data-centric techniques (e.g., data augmentation Yao et al. (2022); Noohdani et al. (2024)) and representation learning Zhang et al. (2022a); Lv et al. (2022); Eastwood et al. (2023); Venkataramani et al. (2024) can effectively reduce the influence of spurious correlations Ye et al. (2024).

To improve memory quality, we propose a simple yet effective sample selection strategy inspired by Toneva et al. (2018), who analyzed the learning dynamics of neural networks and introduced the concept of forgetting events. We populate the buffer with consistently learned (i.e., frequently correctly classified) samples from early training epochs while maintaining class- and task-balanced distributions. This selection is performed directly on the main continual learner and incurs negligible overhead.

The main contributions of this work are as follows:

- To the best of our knowledge, this is the first study to ~~show that the performance of rehearsal-based CL methods degrades significantly under distributional shift, where the i.i.d. assumption no longer holds~~ **systematically evaluate OOD generalization in rehearsal-based CL and to show that state-of-the-art rehearsal-based methods, although designed to prevent forgetting under standard i.i.d. evaluation, degrade significantly under label-preserving covariate shifts at test time**.

- We identify ~~key~~ **three** sources of *overfitting* in buffer-based CL—**unbalanced gradient propagation, spurious correlations within individual training stages, and buffer distribution bias—and argue that each contributes to the erosion of OOD-relevant structure across the task sequence, not only to standard forgetting. Building on this analysis,** ~~and~~ **we** propose AA-RR, a lightweight framework that combines adaptive class-aware reweighting, targeted data augmentation, and representative sample selection.

- We demonstrate the effectiveness and efficiency of AA-RR **for both i.i.d. and OOD performance** through comprehensive experiments on standard CL benchmarks, **achieving the largest gains under covariate shift and** outperforming state-of-the-art baselines.

## 2 Related Work

**Rehearsal-based CL Methods.**  By continually revisiting past experiences drawn from a memory buffer, rehearsal-based methods Lopez-Paz & Ranzato (2017); Chaudhry et al. (2018b; 2019); Aljundi et al. (2019); Prabhu et al. (2020); Sun et al. (2022; 2023); Lin et al. (2023); Huang et al. (2023); Raghavan et al. (2024);

Tong et al. (2025) reinforce earlier knowledge and improve long-term retention across tasks. They achieve this by storing previously seen samples and periodically updating the model with a mixture of old and new data, effectively mitigating catastrophic forgetting.

**Non-Policy-Driven Replay Methods.** GEM Lopez-Paz & Ranzato (2017) mitigates forgetting by constraining gradients to avoid increasing loss on past tasks. A-GEM Chaudhry et al. (2018b) simplifies GEM by projecting gradients only when conflicting with a reference gradient from the buffer. Both methods maintain an equal number of samples for each task. GDUMB Prabhu et al. (2020) adopts a simple strategy by storing samples in the buffer and discarding the model during training. A fresh model is trained from scratch on the buffer at test time. Despite its simplicity, GDUMB often outperforms more complex methods, highlighting the importance of buffer composition over continual updates. ER Chaudhry et al. (2019) is a simple rehearsal approach that maintains the memory buffer via reservoir sampling and retrieves samples uniformly at random. Despite its minimal design, it continues to serve as a strong baseline.

**Policy-Driven Replay Methods.** GSS Aljundi et al. (2019) formulates buffer population as a constraint-reduction problem, selecting samples to minimize gradient interference by maximizing diversity in gradient space. MetaSP Sun et al. (2022) examines the example-level influence on the stability–plasticity trade-off in CL. It computes how each sample affects both remembering past tasks (stability) and learning new ones (plasticity) using influence functions, then fuses these influences via a Pareto-optimal formulation to guide replay buffer selection. SOIF Sun et al. (2023) extends MetaSP by modeling second-order influence—capturing how one sample affects future selections—and introduces a regularizer to limit cumulative bias in the buffer. CSReL Tong et al. (2025) formulates the memory buffer population as a coreset-selection problem via a bilevel formulation. It introduces the concept of reducible loss (ReL) to quantify how much performance would improve if a given sample were added to the buffer, thereby selecting samples that yield the greatest model gain. The CSReL-CL-Prv variant performs per-task selection while reducing task interference using memory, and is referred to as CSReL throughout the paper.

**Reweighting-based Replay Methods.** PCR Lin et al. (2023) introduces a proxy-based contrastive replay strategy, replacing anchor-to-sample pairs with anchor-to-proxy pairs in the contrastive loss to alleviate class-imbalance bias. It employs reservoir sampling to update the buffer.

DELTA Raghavan et al. (2024) addresses long-tailed online CL by decoupling representation learning and classification. It uses contrastive learning for robust features and then fine-tunes the classifier with an equalization loss to balance gradients for underrepresented classes. This method also uses reservoir sampling.

ER-LAS Huang et al. (2023) addresses class imbalance by adjusting logits based on class priors. It demonstrates that Bayes-optimal classification can be approached through a simple logit-adjustment term, with minimal computational cost. Buffer updates are performed using reservoir sampling.

**Forgetting in Stationary Learning Settings.** Toneva et al. (2018) investigates the phenomenon of sample-level forgetting events during standard single-task training: an example is considered forgotten when it goes from correctly classified to incorrectly classified during training. They find that some examples are consistently forgotten while others are never forgotten, and that this distinction is stable across architectures.

## 3 Problem Statement and Analysis

### 3.1 Preliminaries

We consider CL in an offline, buffer-based setting, where a model learns from a sequence of $T$ tasks $\{\mathcal{D}_1, \mathcal{D}_2, \ldots, \mathcal{D}_T\}$. Each task $\mathcal{D}_t = \{(x_i, y_i)\}_{i=1}^{N_t}$ consists of labeled samples from a disjoint class set $\mathcal{C}_t$, i.e., $\mathcal{C}_i \cap \mathcal{C}_j = \emptyset$ for all $i \neq j$. We denote by $\mathcal{C}_{1:t} = \bigcup_{i=1}^{t} \mathcal{C}_i$ the cumulative set of classes observed up to task $t$, and $\mathcal{C}_{<t} = \mathcal{C}_{1:t-1}$ the set of classes seen before task $t$.

The model comprises a feature encoder $f_\theta : \mathbb{R}^d \rightarrow \mathbb{R}^m$ and a linear classifier with class weights $\{w_c \in \mathbb{R}^m\}_{c \in \mathcal{C}_{1:t}}$. For an input $x$, the encoder produces an embedding $z = f_\theta(x)$, and the classifier computes logits

$\ell_c = w_c^\top z$. The corresponding probabilities are obtained via a softmax:

$$p_c = \frac{e^{(\ell_c)}}{\sum_{j \in \mathcal{C}_{1:t}} e^{(\ell_j)}}. \tag{1}$$

During training on task $t$, the model has access to the full dataset $\mathcal{D}_t$ and a memory buffer $\mathcal{B}$ that stores exemplars from past tasks. The training objective minimizes the expected cross-entropy loss over both sources:

$$\mathcal{L} = \mathbb{E}_{(x,y) \sim \mathcal{D}_t \cup \mathcal{B}} \big[ - \log p_y \big]. \tag{2}$$

In this offline setup, multiple passes over $\mathcal{D}_t$ are allowed, but previous data $\mathcal{D}_{<t}$ are only accessible via $\mathcal{B}$.

## 3.2 Unbalanced Gradient Propagation

**The gradient analysis in this section characterizes a standard CL mechanism: imbalance between current-task data and replayed past-task data (Caccia et al., 2021; Lin et al., 2023; Huang et al., 2023). By itself, this analysis explains forgetting, classifier bias, and representation drift under the training distribution. We use it as a foundation for our OOD argument. Specifically, if representations or class boundaries learned in earlier tasks contain features that support robustness under covariate shift, gradient imbalance, and feature drift can erode these features over time.**

Cross-entropy training introduces a gradient imbalance between new and old tasks, primarily due to the disproportionate number of samples from $\mathcal{D}_t$ compared to the buffer $\mathcal{B}$. For a sample $(x, y)$ with embedding $z$, the gradient with respect to the classifier weights is:

$$\frac{\partial \mathcal{L}}{\partial w_c} = (p_c - \mathbf{1}[c = y]) \, z. \tag{3}$$

Where, $\mathbf{1}[.]$ is the indicator function. Meanwhile, the gradient with respect to the feature embedding is:

$$\frac{\partial \mathcal{L}}{\partial z} = \sum_{j \in \mathcal{C}_{1:t}} p_j w_j - w_y. \tag{4}$$

These updates pull $w_y$ and $z$ toward each other while pushing them away from other class weights, improving discrimination for frequent classes.

Let $n_t = |\mathcal{D}_t|$ and $n_b = |\mathcal{B}|$. The expected gradient on a weight $w_c$ can be decomposed as:

$$\mathbb{E}_{\mathcal{D}_t \cup \mathcal{B}} \left[ \frac{\partial \mathcal{L}}{\partial w_c} \right] = \frac{n_t}{n_t + n_b} \mathbb{E}_{\mathcal{D}_t} \left[ \frac{\partial \mathcal{L}}{\partial w_c} \right] + \frac{n_b}{n_t + n_b} \mathbb{E}_{\mathcal{B}} \left[ \frac{\partial \mathcal{L}}{\partial w_c} \right]. \tag{5}$$

Since $n_t \gg n_b$, gradients from $\mathcal{D}_t$ dominate. Consequently, weights of new classes receive frequent, large updates of $w_c^{(t+1)} \approx w_c^{(t)} - \eta \sum_{(x,y=c) \in \mathcal{D}_t} (p_c - 1) z$, whereas old-class weights are updated sparsely through replay, i.e. $w_c^{(t+1)} \approx w_c^{(t)} - \eta \sum_{(x,y=c) \in \mathcal{B}} (p_c - 1) z$.

**Gradient Polarity Across Tasks.** The imbalance also extends to non-target classes. For any sample $(x, y)$, the gradient with respect to the logit of class $c \neq y$ is $\partial \mathcal{L} / \partial \ell_c = p_c$, implying that each non-target weight receives a repulsive update $\partial \mathcal{L} / \partial w_c = p_c z$.

When $x \sim \mathcal{D}_t$ (new data), all old classes $c \in \mathcal{C}_{<t}$ appear only as non-targets, accumulating strong negative gradients. Conversely, when $x \sim \mathcal{B}$ (replayed data), new classes $c \in \mathcal{C}_t$ act as non-targets, but their logits are low and the induced gradients are weak, $\big\| \frac{\partial \mathcal{L}}{\partial w_{c_{\text{old}}}} \big\|_{x \sim \mathcal{D}_t} \gg \big\| \frac{\partial \mathcal{L}}{\partial w_{c_{\text{new}}}} \big\|_{x \sim \mathcal{B}}$. Hence, old-class weights are continually pushed away from evolving feature distributions, amplifying forgetting and representation drift, $\langle f_\theta(x_{\text{old}}), w_{c_{\text{old}}} \rangle \downarrow, \quad \langle f_\theta(x_{\text{old}}), w_{c_{\text{new}}} \rangle \uparrow$.

**This asymmetric polarity compounds over the task sequence. A class $c \in \mathcal{C}_{<t}$ introduced at task $t_c$ has, by task $t$, been a past class over a span of training stages that grows with**

its age $t - t_c$. **During each such stage, the class appears predominantly as a non-target in current-task gradient computations and only sparsely as a target through buffer replay. Older classes—corresponding to larger $t - t_c$—therefore accumulate more rounds of this drift-inducing update pattern than recently introduced past classes. We use this age-dependent exposure as one motivation for the recency-aware term in the adaptive reweighting rule introduced in Section 4.1.**

With a fixed buffer size $|\mathcal{B}|$, the number of exemplars per old class decreases as tasks accumulate, $|\mathcal{B}_c| \approx \frac{|\mathcal{B}|}{|\mathcal{C}_{<t}|}$, $c \in \mathcal{C}_{<t}$, This progressively weakens the gradient signal from past tasks, amplifying both classifier bias and feature drift. **Crucially, this drift is not selective with respect to the kind of information being lost: it erodes whatever structure was learned for old classes, so any robustness to covariate shift acquired in earlier stages is lost together with the i.i.d. content of past tasks. This is the bridge from the gradient-imbalance analysis above to the OOD-specific design choices in Section 4.**

**The fixed buffer also creates a per-example exposure imbalance under multi-epoch training. Each epoch iterates over the current-task data, while memory samples are drawn repeatedly during gradient updates. Since the buffer is much smaller than the current-task dataset, individual buffered samples can be replayed multiple times per epoch, whereas each current-task sample is typically seen once per epoch. Past classes are therefore represented by a small set of exemplars that receive repeated exposure, increasing the risk that the model memorizes those specific buffered samples rather than learning features that generalize for the corresponding classes.**

### From Gradient Imbalance to OOD Degradation: A Margin-Based View

**The gradient-imbalance analysis in Section 3.2 describes a CL-induced optimization mechanism that affects both i.i.d. and OOD performance, but does not by itself require an OOD assumption. We now make the connection to OOD degradation explicit through a margin-based argument: gradient-induced margin compression for past classes affects clean and corrupted inputs asymmetrically, and is therefore a natural mechanism for the disproportionately large OOD drops observed in Figure 1 and Table 1.**

**For a sample $(x, y)$ with feature $z = f_\theta(x)$ and classifier weights $\{w_c\}_{c \in C_{1:t}}$, define the classification margin as**

$$m(x) = w_y^\top f_\theta(x) - \max_{c \neq y} w_c^\top f_\theta(x).$$

**The prediction at $x$ equals $y$ whenever $m(x) > 0$, so i.i.d. 0-1 accuracy depends only on the sign of $m(x)$.**

**For a label-preserving corruption $q$, as used in our OOD evaluation (Section 5.1), we assume the induced feature shift is bounded by**

$$\big\| f_\theta(q(x)) - f_\theta(x) \big\| \leq L_\theta \, \delta,$$

**where $L_\theta$ captures the local sensitivity of the representation under $\theta$ and $\delta$ measures the corruption magnitude. Writing $\Delta = f_\theta(q(x)) - f_\theta(x)$ and letting $c^\star = \arg\max_{c \neq y} w_c^\top f_\theta(q(x))$, we have**

$$m(q(x)) = (w_y - w_{c^\star})^\top f_\theta(x) + (w_y - w_{c^\star})^\top \Delta \geq m(x) - \max_{c \neq y} \|w_y - w_c\| \cdot L_\theta \, \delta,$$

**using $(w_y - w_{c^\star})^\top f_\theta(x) \geq m(x)$—which follows from $w_{c^\star}^\top f_\theta(x) \leq \max_{c \neq y} w_c^\top f_\theta(x)$—and using Cauchy–Schwarz together with $\|w_y - w_{c^\star}\| \leq \max_{c \neq y} \|w_y - w_c\|$ on the second term. Therefore, a sufficient condition for the prediction to be preserved under $q$ is**

$$m(x) > \max_{c \neq y} \|w_y - w_c\| \cdot L_\theta \, \delta.$$

This condition makes the i.i.d./OOD asymmetry explicit. Preserving clean correctness at $x$ requires $m(x) > 0$; preserving correctness under label-preserving corruption requires $m(x)$ to be large enough to absorb the feature displacement induced by $q$. Two models with comparable i.i.d. 0-1 accuracy can therefore differ sharply in OOD accuracy if one has more compressed margins on the relevant samples.

**Connection to Section 3.2.** The analysis of Section 3.2 explains why margin compression naturally arises for past classes in rehearsal-based CL. Past-class weights $w_{c_{\mathrm{old}}}$ receive only sparse positive updates through the limited buffer, while they appear repeatedly as non-target classes during current-task training and accumulate the repulsive gradients identified in the gradient-polarity analysis. At the same time, the encoder is primarily driven by current-task data, producing the feature drift $\langle f_\theta(x_{\mathbf{old}}), w_{c_{\mathrm{old}}} \rangle \downarrow$, $\langle f_\theta(x_{\mathbf{old}}), w_{c_{\mathrm{new}}} \rangle \uparrow$. Over the task sequence, these two effects act jointly on the margin $m(x)$ for past-class samples. As training proceeds beyond task $t_c$, the target term $w_y^\top f_\theta(x_{\mathbf{old}})$ tends to decrease (following the feature drift above), while the maximum competing term $\max_{c \neq y} w_c^\top f_\theta(x_{\mathbf{old}})$ tends to be driven up by new-class weights $c \in C_t$ that have been updated toward the evolving feature distribution. The net effect is that $m(x_{\mathbf{old}})$ typically contracts as more tasks are observed. Under i.i.d. evaluation, this contraction becomes visible in 0-1 accuracy only once the margin crosses zero. Under OOD evaluation with label-preserving corruptions, however, even positive but small margins can fail the sufficient condition above ($m(x) > \max_{c \neq y} \|w_y - w_c\| \cdot L_\theta \, \delta$), giving a disproportionately larger drop in OOD accuracy than in i.i.d. accuracy—consistent with the empirical pattern in Figure 1 and Table 1.

**Implications for AA-RR.** Reading condition $m(x) > \max_{c \neq y} \|w_y - w_c\| \cdot L_\theta \, \delta$ component-wise clarifies how each component of AA-RR (Section 4) contributes to OOD robustness in addition to mitigating standard forgetting. The adaptive reweighting mechanism (Section 4.1) directly counteracts the gradient imbalance that compresses old-class margins by modulating the effective gradients through $\alpha_c$ as discussed in Section 4.1. The augmentation pipeline in Section 4.2 encourages representations that are less sensitive to label-preserving perturbations, corresponding to a smaller effective $L_\theta$ in the condition above, which loosens the requirement on $m(x)$ for the prediction to be preserved. The correctness-guided buffer management (Section 4.3) retains consistently learned samples under class- and task-balance constraints; while this criterion does not target margins directly, it stabilizes the replay signal for each past class and prevents the buffer from being dominated by outliers or recently observed classes, both of which reduce the severity of the gradient imbalance analyzed in Section 3.2 and therefore indirectly help preserve past-class margins across the task sequence.

### 3.3 Logit Reweighting via Class-Prior Adjustment

Motivated by the gradient imbalance and margin-compression effects discussed above, ~~A~~a common strategy ~~to mitigate this unbalanced gradient propagation~~ is to reweight the softmax cross-entropy loss based on class frequency. Recent methods such as PCR Lin et al. (2023), DELTA Raghavan et al. (2024), and ER-LAS Huang et al. (2023) adopt this approach by estimating class priors using recently observed classes and scaling each class's loss contribution proportional to its observed frequency.

**Proxy-based Contrastive Replay (PCR).** PCR Lin et al. (2023) addresses gradient imbalance by replacing anchor-to-sample relationships in the contrastive loss with anchor-to-proxy relationships. Specifically, for a training sample $(x, y)$, the loss is defined as:

$$\mathcal{L}_{\mathrm{PCR}} = -\frac{1}{|P(x)|} \sum_{p \in P(x)} \log \frac{e^{\left(\cos(\tilde{w}_p^\top \tilde{z})/\gamma \;\; \mathrm{sim}(\tilde{w}_p, \tilde{z})/\tau\right)}}{\sum_{c \in C_B} e^{\left(\cos(\tilde{w}_c^\top \tilde{z})/\gamma \;\; \mathrm{sim}(\tilde{w}_c, \tilde{z})/\tau\right)}}, \tag{6}$$

where ~~cos(·)~~ $\text{sim}(a,b) = \frac{a^\top b}{\|a\|\,\|b\|}$ denotes cosine similarity **between two vectors (which reduces to the inner product $\tilde{a}^\top \tilde{b}$ for $\ell_2$-normalized inputs)**, $\tilde{z}$ and $\tilde{w}_c$ are $\ell_2$-normalized representations and class proxies, respectively, $\tau$ is the temperature parameter, and $C_B$ denotes the set of classes present in the current batch (allowing repetitions). **We use the notation $\text{sim}(\cdot,\cdot)$ consistently throughout the paper.** Assuming $n_c$ denotes the frequency of class $c$ in the current batch, the loss simplifies to:

$$\mathcal{L}_{\text{PCR}} = -\log \frac{e^{\left(\overline{\cos(\tilde{w}_y^\top z)/\tau}\ \text{sim}(\tilde{w}_y, \tilde{z})/\tau\right)}}{\sum_{c\in\mathcal{C}_{1:t}} e^{\left(\overline{\cos(\tilde{w}_c^\top z)/\tau}\ \text{sim}(\tilde{w}_c, \tilde{z})/\tau\right)} \cdot n_c}. \tag{7}$$

**DELTA.** DELTA Raghavan et al. (2024) addresses long-tailed *online* CL by decoupling representation learning and classifier optimization. It adopts a two-stage training scheme: the encoder is first trained using a supervised contrastive loss to learn class-discriminative features, followed by a classifier fine-tuning stage using an *Equalization Loss*:

$$\mathcal{L}_{\text{EQ}} = -\log \frac{e^{(\ell_y + \log \pi(y))}}{\sum_{c\in\mathcal{C}_{1:t}} e^{(\ell_c + \log \pi(c))}}, \tag{8}$$

where $\ell_c$ denotes the logit for class $c$, and $\pi(c)$ is the estimated prior probability of class $c$, computed from the data observed so far up to task $t$ in the training stream.

**Logit-Adjusted Softmax (LAS).** LAS Huang et al. (2023) performs logit adjustment using estimated class priors $\pi(c)$ computed over a sliding window of recent training batches, along with a temperature parameter $\tau$:

$$\mathcal{L}_{\text{LAS}} = -\log \frac{e^{(\ell_y + \tau \log \pi(y))}}{\sum_{c\in\mathcal{C}_{1:t}} e^{(\ell_c + \tau \log \pi(c))}}. \tag{9}$$

Rather than maintaining cumulative statistics over the full data stream (as done in DELTA), LAS estimates $\pi(c)$ from a sliding window that tracks class frequencies over the most recent $k$ batches.

### 3.4 Limitations and Motivation

Class frequency-based reweighting methods—such as DELTA, LAS, and PCR—aim to mitigate gradient imbalance by adjusting logits or loss contributions based on estimated class priors or class count. LAS and PCR use batch-local or sliding-window statistics, making them highly sensitive to short-term sampling fluctuations. When a class is missing from recent batches, its estimated frequency drops to zero, eliminating its softmax contribution and halting further learning. DELTA addresses this volatility by maintaining cumulative class priors over the entire data stream, improving stability. However, it neglects recency: two classes with identical cumulative counts may differ significantly in how recently they were observed. All three methods—DELTA, LAS, and PCR—treat class frequency as temporally static, failing to adapt reweighting based on how stale or current a class is in the buffer.

To overcome these limitations **—which affect both i.i.d. and OOD performance, since unstable old-class representations cannot be expected to retain robustness to covariate shift—** ~~,~~ we propose AA-RR**, which can be viewed as extending prior-based logit adjustment to the CL setting. Instead of estimating a class prior only from batch-local, sliding-window, or cumulative class frequencies, AA-RR constructs a temporally adaptive prior-like correction from CL statistics.** ~~, a method that~~ **AA-RR** explicitly regulates gradient propagation to ensure consistent, temporally adaptive, and balanced updates across all classes in CL**, and pairs this regulation with augmentation and selection components targeted specifically at OOD robustness (Sections 4.2 and 4.3). AA-RR uses a structured, CL-motivated adaptive reweighting rule designed to incorporate task progression, class recency, and replay-buffer frequency, rather than relying only on static or short-window class counts.**

### 3.5 Relation to Traditional OOD Generalization

The OOD-degradation mechanism described in Section "From Gradient Imbalance to OOD Degradation: A Margin-Based View" (*note: for the camera-ready version, we will make it a cross-reference*) is closely related to several established lines of work in stationary OOD generalization, although the underlying setting differs in important ways. Standard OOD methods typically assume a fixed training set and address test-time distribution shift through strategies such as (i) importance weighting under covariate shift (Shimodaira, 2000; Sugiyama et al., 2007); (ii) robust data augmentation and vicinal risk minimization (Zhang et al., 2017; Hendrycks et al., 2019; Yao et al., 2022); (iii) representation learning (Zhang et al., 2022a; Lv et al., 2022; Eastwood et al., 2023); or (iv) group-distributionally-robust optimization (Sagawa et al., 2019). Related but distinct from these OOD methods, long-tail and class-imbalance methods use class reweighting or distributional robustness losses to reduce bias toward frequent classes (Cui et al., 2019; Samuel & Chechik, 2021). The rehearsal-based CL setting we consider differs from both in three respects: the training distribution changes over time, past data are accessible only through a limited replay buffer, and individual buffered samples can be replayed repeatedly across epochs, and often multiple times within an epoch when the buffer is small relative to the current task. As a consequence, the OOD failures observed in Figure 1 and Table 1 are not explained by test-time covariate shift alone—they are also shaped by CL-specific training dynamics, namely buffer dilution, class- and task-recency imbalance, and overfitting to repeatedly replayed exemplars (Section 3.2).

Each component of AA-RR can be read as an adaptation of one of these strategies to the CL setting: 1. The adaptive reweighting in Section 4.1 draws on both covariate-shift weighting and class-imbalance correction, but is distinct from either. Like covariate-shift importance weighting, it modifies the effective training distribution, but it does not estimate a target-domain density ratio. Like long-tail class reweighting, it corrects class imbalance, but the imbalance it targets is the *effective training prior* induced by CL-specific statistics—task progression, class age, and per-class replay-buffer population—rather than a static empirical class-frequency distribution. The prior-adjustment interpretation in Section 4.1 makes this connection precise. 2. The augmentation pipeline in Section 4.2 is related to OOD-robustness methods that encourage invariance to label-preserving low-level transformations, but is applied within a replay loop, where it must diversify both current-task batches and the much smaller, frequently replayed memory. 3. The correctness-guided buffer selection in Section 4.3 builds on the idea of sample-forgetting statistics tracked during stationary training (Toneva et al., 2018), but is designed for a fixed-capacity buffer that must be rebalanced across both classes and tasks at every reallocation under non-stationary task arrival.

Two ingredients common in the stationary OOD literature are not included as standalone components of AA-RR: an explicit invariant-representation regularizer (Zbontar et al., 2021; Lv et al., 2022) and a group-distributionally-robust objective (Sagawa et al., 2019). In Section 5.2, we report a direct empirical comparison against rehearsal-based instantiations of four alternatives, including strong augmentation (Khosla et al., 2020) alone, a Barlow-Twins-style factorization regularizer (Zbontar et al., 2021; Lv et al., 2022), Mixup (Zhang et al., 2017), and task-level GroupDRO (Sagawa et al., 2019). The results show that none of the tested variants matches AA-RR's OOD performance.

## 4 Proposed Method

We propose Adaptive Augmented Reweighted Replay (AA-RR), a lightweight framework designed to improve ~~generalization and robustness~~ standard i.i.d. retention while primarily targeting OOD robustness in rehearsal-based CL. AA-RR addresses the challenges of gradient imbalance, spurious correlation, and buffer distribution bias, which depend on which samples are retained and whether class and task balance are preserved. It does so through three core components, each tied to a distinct mechanism: 1. an adaptive class-aware loss reweighting mechanism, which preserves past-task representations under

**non-stationary training and is a prerequisite for retaining any structure learned earlier, including OOD-robust structure**; 2. targeted data augmentation ~~to mitigate~~ **, which addresses OOD robustness directly by reducing reliance on** spurious, **task-specific** correlations; and 3. a representative sample selection strategy ~~to improve memory quality~~**, which ensures that the retained samples are class- and task-balanced and consistently learned, stabilizing the representations that the reweighting term is asked to preserve**.

In this section, we first describe the core formulation of our adaptive reweighting strategy, which directly addresses the gradient imbalance and recency bias inherent in buffer-based CL. We then introduce the augmentation and selection components, which further enhance ~~generalization~~ **robustness and retention** by reducing ~~overfitting~~ **reliance on spurious correlations** and improving buffer representativeness.

The overall AA-RR training procedure is summarized in Algorithm 1 in Appendix A.1, illustrating how these components are integrated into CL.

## 4.1 Adaptive Class-Aware Reweighting

We modify the standard objective in Eq. 2 by introducing a per-class scaling factor $\alpha_c > 0$, applied inside the softmax. For a training sample $(x_i, y_i)$ with logits $\ell_{i,c}$, we compute the reweighted softmax cross-entropy loss as:

$$\mathcal{L}_{\text{AA-RR}} = \mathbb{E}_{(x,y)\sim\mathcal{D}_t\cup\mathcal{B}}\left[-\log\frac{e^{(\ell_y)}\cdot\alpha_y}{\sum_{j\in\mathcal{C}_{1:t}}e^{(\ell_j)}\cdot\alpha_j}\right]. \tag{10}$$

When all $\alpha_c = 1$, this reduces to the standard cross-entropy. Otherwise, $\alpha_c$ modifies both the softmax distribution and the gradient magnitude, correcting for class imbalance at training time.

**Equivalently, Eq. 10 can be written as a logit-adjusted softmax loss:**

$$\mathcal{L}_{\text{AA-RR}} = \mathbb{E}_{(x,y)\sim\mathcal{D}_t\cup\mathcal{B}}\left[-\log\frac{e^{(\ell_y+\log\alpha_y)}}{\sum_{j\in\mathcal{C}_{1:t}}e^{(\ell_j+\log\alpha_j)}}\right].$$

**Thus, $\alpha_c$ acts as an unnormalized, class-dependent prior correction, or equivalently, $\log\alpha_c$ acts as a class-dependent logit offset. If desired, one may define a normalized effective prior $\tilde{\pi}_c = \alpha_c/\sum_{j\in\mathcal{C}_{1:t}}\alpha_j$; the common normalization constant cancels inside the softmax.**

**Dynamic Per-Class Weighting.** The ~~class~~ **prior-like correction** weights $\{\alpha_c\}$ are computed dynamically at each training iteration **using observable statistics from the CL process. The resulting rule is not intended as a uniquely optimal weighting formula; rather, it is a compact empirical parameterization of three CL-specific imbalance factors identified in Section 3.2**. Let $t$ be the current task index, and $\mathcal{C}_t$ the set of classes introduced in task $t$. For each class $c$, we define:

$$\alpha_c = \left(\frac{1}{t-1}\right)^{\gamma_1}\cdot\left(\frac{1}{t-t_c}\right)^{\gamma_2}\cdot\left(\frac{f_c}{\bar{f}_{\mathcal{C}_t}+\hat{n}}\right)^{\gamma_3} \tag{11}$$

We apply Eq. 11 to all classes $c \in \mathcal{C}_{1:t}$, with the understanding that for current-task classes $c \in \mathcal{C}_t$, the reweighting terms are ignored and we set $\alpha_c = 1$. This defines the current task as the reference point, with no decay or correction applied.

For all previous-task classes $c \in \mathcal{C}_{<t}$, the three multiplicative terms in Eq. 11 serve distinct purposes. Where, $t_c$ is the task number in which class $c$ was introduced; $f_c$ is the number of buffer samples for class $c$; $\bar{f}_{\mathcal{C}_t}$ is the average buffer count for current-task classes; $\hat{n} = \frac{n_{\text{task}}}{|\mathcal{C}_t|}$ is the expected per-class sample count for $\mathcal{D}_t$; $\gamma_1, \gamma_2, \gamma_3 \in \mathbb{R}_+$ are tuning parameters.

The three multiplicative terms address distinct aspects of training imbalance. **Term 1** $\left(\frac{1}{t-1}\right)^{\gamma_1}$ counteracts the growing disparity between buffered and current-task samples as training progresses. It adaptively amplifies the contribution of earlier-task classes by scaling their weights according to their diminishing presence in the training distribution, thus mitigating class imbalance caused by buffer dilution. In effect, this

term addresses the imbalance between current and previous tasks. **Term 2** $\left(\frac{1}{t-t_c}\right)^{\gamma_2}$ applies a temporal decay based on the age of class $c$, giving higher priority to older classes at greater risk of forgetting. **Term 3** $\left(\frac{f_c}{\bar{f}_{c_t}+\hat{n}}\right)^{\gamma_3}$ adjusts for repeated exposure of buffer samples. As training progresses, buffer samples are seen more frequently than current-task samples, which can cause the model to overfit them and lead to overconfident predictions.

**Connection to Section 3.2 and Scope of the Formulation.** Each of the three terms in Eq. 11 corresponds to an imbalance factor discussed in Section 3.2. Term 1, $(1/(t-1))^{\gamma_1}$, captures *buffer dilution*: under a fixed buffer size, the per-past-task quota of buffered exemplars is approximately $|\mathcal{B}|/(t-1)$ and shrinks as the task index $t$ grows. Term 2, $(1/(t-t_c))^{\gamma_2}$, captures *class age*: classes introduced earlier—corresponding to larger $t-t_c$—have been past classes over more subsequent training stages and therefore subjected to more rounds of the asymmetric, drift-inducing update pattern. Term 3, $(f_c/(\bar{f}_{c_t}+\hat{n}))^{\gamma_3}$, captures the relative per-class *population* of class $c$ in the replay buffer compared with the effective current-task class count.

Because $\alpha_c$ is applied inside the softmax in Eq. 10 rather than as an external loss multiplier, smaller values of $\alpha_c$ for past classes act as a logit adjustment with two distinct effects. For current-task samples, reducing $\alpha_c$ on an old non-target class decreases its softmax probability and therefore reduces the repulsive gradient applied to its classifier weight $w_c$, mitigating drift. For buffer-replay samples in which an old class is the target, a smaller target-side $\alpha_y$ reduces $p_y$ and can amplify the corrective gradient on $w_y$, strengthening the replay signal when the model is under-confident on the past class. In this way, Eq. 11 uses task progression, class age, and per-class buffer population to modulate the effective gradients received by past classes—a structured response to the imbalance factors identified in Section 3.2.

In log space, it becomes an additive combination of the three correction signals,

$$\log \alpha_c = \gamma_1 \log \frac{1}{t-1} + \gamma_2 \log \frac{1}{t-t_c} + \gamma_3 \log \frac{f_c}{\bar{f}_{c_t}+\hat{n}},$$

with $\gamma_1, \gamma_2, \gamma_3$ controlling the relative strength of each correction. In principle, one could optimize a separate $\alpha_c$ for each class or task directly—for instance, via Bayesian optimization—but doing so would introduce at least $|\mathcal{C}_{1:T}|$ degrees of freedom, scale validation cost with the number of classes, and risk overfitting to a specific task sequence. By contrast, our formulation uses only three global hyperparameters, computes $\alpha_c$ dynamically from observable training statistics $(t, t_c, f_c, \bar{f}_{\mathcal{C}_t}, \hat{n})$, and—as we show in Section 5.3 and Appendix A.4—yields performance that is stable across a wide range of these hyperparameters under both i.i.d. and OOD evaluation.

**Prior-Adjustment Interpretation.** Writing the loss in Eq. 10 in additive logit-adjustment form shows that $\log \alpha_c$ plays the same structural role as the prior term $\log \pi(c)$ in DELTA's Equalization Loss (Eq. 8) and ER-LAS's Logit-Adjusted Softmax (Eq. 9): it acts as an additive shift on the logit of class $c$ at training time, modulating the effective per-class learning pressure. Our reweighting can therefore be read as a *temporally adaptive prior correction*. Where DELTA estimates $\pi(c)$ from cumulative class counts over the whole data stream, and LAS estimates it from a sliding window of recent batches, $\alpha_c$ replaces this purely frequency-based prior with one that depends on three observable CL signals: per-task buffer share $(1/(t-1))$, class age $(1/(t-t_c))$, and per-class buffer population relative to the effective current-task class size $(f_c/(\bar{f}_{\mathcal{C}_t}+\hat{n}))$. AA-RR therefore retains the prior-adjustment view that LAS and DELTA make explicit, but generalizes the prior from a static or batch-local frequency to one that tracks the temporal structure of the task stream and the state of the buffer. The limitations of frequency-only priors that we discussed in Section 3.4—sensitivity to short-term sampling fluctuations (LAS, PCR), and insensitivity to recency (DELTA)—are addressed directly by this generalization.

In sum, our reweighted softmax loss provides a ~~principled yet~~ structured, lightweight, **and empirically motivated** mechanism—**guided by the gradient-imbalance analysis in Section 3.2**—to balance class

contributions during training. It mitigates ~~the~~ overfitting ~~of~~ **to** recent classes and preserves performance on earlier tasks~~—~~ without requiring architectural changes or auxiliary components **and optimization**.

## 4.2 Augmentation for OOD Robustness

A key insight from recent data-centric studies Yao et al. (2022); Noohdani et al. (2024) is that enhancing input diversity through strong, semantically preserving augmentations can substantially improve model robustness and reduce overfitting to spurious correlations. Following this principle, AA-RR adopts a data-centric augmentation approach to improve generalization under covariate shift in CL. By enriching the effective training distribution, the model becomes less sensitive to superficial, task-specific patterns and better equipped to handle OOD inputs. **In the margin view of Section "From Gradient Imbalance to OOD Degradation: A Margin-Based View" (*note: for the camera-ready version, we will make it a cross-reference*), such augmentations can be interpreted as reducing the effective representation sensitivity $L_\theta$ to label-preserving perturbations, thereby making the OOD margin condition easier to satisfy.**

Motivated by this perspective, we employ a strong augmentation pipeline inspired by Supervised Contrastive Learning (SupCon) Khosla et al. (2020). The pipeline includes transformations such as random cropping, horizontal flipping, color jittering, and grayscaling—augmentations known to produce diverse yet label-consistent views. These augmentations encourage the model to learn representations that are invariant to low-level appearance changes while preserving class-discriminative information.

For each training sample, we generate an augmented view using this pipeline and concatenate it with the original image in the training batch. To further mimic contrastive learning behavior, we replace the standard dot product in the softmax classifier with cosine similarity $\sim(\tilde{w}_c, \tilde{z})$ **defined in Section 3.3,** and introduce a temperature parameter $\tau$ to scale the logits. This yields the reweighted softmax cross-entropy objective applied to the combined batch:

$$\mathcal{L}_{\text{AA-RR}} = \mathbb{E}_{(x,y)\sim\mathcal{D}_t\cup\mathcal{B}} \left[ -\log \frac{e^{\left(\cancel{\cos(\tilde{w}_y^\top\tilde{z})/\tau}\ \sim(\tilde{w}_y,\tilde{z})/\tau\right)} \cdot \alpha_y}{\sum_{j\in\mathcal{C}_{1:t}} e^{\left(\cancel{\cos(\tilde{w}_j^\top\tilde{z})/\tau}\ \sim(\tilde{w}_j,\tilde{z})/\tau\right)} \cdot \alpha_j} \right] \tag{12}$$

Here, $\tilde{z}$ is the $\ell_2$-normalized feature representation of the input, $\tilde{w}_j$ is the normalized class weight vector, $\alpha_j$ is the class reweighting factor (defined in Section 4.1), and $\mathcal{C}_{1:t}$ denotes all classes observed up to task $t$. **As in Eq. 10, the factors $\alpha_j$ can equivalently be interpreted as adding $\log\alpha_j$ to the normalized similarity logits, preserving the same temporally adaptive prior-adjustment view in the augmented training objective.**

This augmentation strategy strengthens representational invariance, expands the effective training distribution, and mitigates overfitting to spurious or task-specific patterns. Consequently, it significantly improves the model's ability to generalize to OOD samples—addressing a core limitation of existing rehearsal-based CL methods.

## 4.3 Correctness-Guided Buffer Management

We introduce a correctness-based strategy for exemplar selection and memory reallocation within the fixed-capacity buffer $\mathcal{B}$. Our approach measures how often each sample is predicted correctly during early training and retains only those samples that are consistently well learned, while ensuring class- and task-balanced retention of informative exemplars across the learning sequence.

**Correctness Scoring.** During training on task $\mathcal{D}_t = \{(x_i, y_i)\}_{i=1}^{N_t}$, we monitor each sample's prediction outcome over the first $E$ epochs. Let $f_\theta^{(e)}$ denote the encoder parameters after epoch $e$, and $w_c$ the classifier weight of class $c \in \mathcal{C}_{1:t}$. The correctness indicator for sample $(x_i, y_i)$ at epoch $e$ is defined as:

$$c_{e,i} = \mathbf{1}\left[\arg\max_{c\in\mathcal{C}_{1:t}} w_c^\top f_\theta^{(e)}(x_i) = y_i\right] \tag{13}$$

The cumulative correctness count of each sample is then, $r_i = \sum_{e=1}^{E} c_{e,i}$, where $r_i \in \{0, \ldots, E\}$ indicates the **early** epochs number in which sample $i$ was classified correctly. **In practice, we never materialize the full $E \times N_t$ matrix of indicators $\{c_{e,i}\}$: $r_i$ is maintained as a single per-sample counter that is updated online in place whenever sample $i$ is encountered during an early epoch, so only $N_t$ scalar counters are held in memory during current-task training.** This score serves as a ranking criterion for buffer selection.

**Task–Class Allocation.** The memory buffer $\mathcal{B}$ maintains a fixed capacity $B$ shared among all observed tasks. After completing task $t$, buffer capacity is distributed evenly across the $t$ tasks, i.e. $m_t = \lfloor \frac{B}{t} \rfloor$, $\sum_{i=1}^{t} m_i = B$. Each task quota $m_t$ is further divided uniformly among its class set $\mathcal{C}_t$, i.e. $m_{t,c} = \lfloor \frac{m_t}{|\mathcal{C}_t|} \rfloor$, for each $c \in \mathcal{C}_t$. This ensures balanced memory allocation both across tasks and within their constituent classes.

**Buffer Update Rule.** At the end of training on $\mathcal{D}_t$, the buffer is updated according to the three steps; **1. Current-task selection:** For each class $c \in \mathcal{C}_t$, rank samples in $\mathcal{D}_t$ by their correctness counts $r_i$ in descending order and ~~select~~ **insert** the top $m_{t,c}$ exemplars **into the buffer in this order**, $\mathcal{S}_{t,c} = \mathrm{Top}_{m_{t,c}}(\{(x_i, y_i) \in \mathcal{D}_t : y_i = c\}, r_i)$. **2. Retention from previous tasks:** For each prior task $i < t$ and class $c \in \mathcal{C}_i$, retain only the ~~top~~ **first** $m_{i,c}$ samples in the ~~existing buffer, ranked by their stored correctness values~~ **per-class buffer ordering established at the end of task $i$. Because samples were inserted in descending order of $r_i$, this insertion order itself ranks them by correctness; the buffer therefore carries the correctness ranking implicitly, and no per-buffer-sample correctness score needs to be stored alongside buffer entries**. **3. Merge and prune:** The updated buffer is obtained by combining the retained and newly selected subsets, $\mathcal{B} = \bigcup_{i=1}^{t} \bigcup_{c \in \mathcal{C}_i} \mathcal{S}_{i,c}$, $|\mathcal{S}_{i,c}| = m_{i,c}$. ~~If any class exceeds its quota $m_{i,c}$, the least-correct samples are removed~~ **Any sample beyond a class's quota is dropped from the tail of its per-class ordering**.

This procedure guarantees that $|\mathcal{B}| = B$ at all times and that each class retains a balanced and correctness-prioritized subset of exemplars. By discarding less reliably learned samples when memory is full, the buffer evolves into a compact and stable repository of well-learned examples, effectively supporting CL without catastrophic forgetting.

**Memory Complexity.** **Compared with a standard rehearsal baseline that stores only the memory buffer (capacity $B$), model parameters, and the usual optimizer states and activations, correctness-guided buffer management introduces only a temporary array of $N_t$ integer counters $\{r_i\}$ during the early epochs of each task. As discussed above, no $E \times N_t$ correctness matrix and no per-buffer-sample correctness score need to be retained: $r_i$ is updated in place, and the buffer encodes the ranking through insertion order alone. On Split CIFAR-100, this corresponds to $N_t = 5000$ counters per task; on Split Mini-ImageNet and Split Tiny-ImageNet, to $N_t = 10000$ counters per task. With $E = 9$ in our default configuration, each counter fits in a single byte, so the additional working memory is approximately 5 KB and 10 KB, respectively—negligible compared to a ResNet-18 backbone ($\approx$ 45 MB at float32), its optimizer states, activations, and the buffer itself. The counter array is also released at the end of each task and does not accumulate across the task sequence.**

## 5 Experiments

### 5.1 Setup

For fair and consistent comparison across CL methods, we extend the Mammoth framework following the experimental protocols described in Buzzega et al. (2020); Boschini et al. (2022). All experiments are conducted on a Quadro RTX 8000 GPU with CUDA 10.2.

**Datasets.** We evaluate all methods on three widely used class-incremental learning benchmarks. Split CIFAR-100 partitions the 100 classes into 10 sequential tasks of 10 classes each, with 32×32 pixel images

Boschini et al. (2022). Split Mini-ImageNet follows a similar format, comprising 5 tasks of 20 classes each, with images resized to 32×32 pixels Sun et al. (2022) (results using the original 84×84 resolution are provided in the ablation study in Section 5.3). Split Tiny-ImageNet includes 200 classes divided into 10 tasks of 20 classes each, also resized to 32×32 pixels Buzzega et al. (2020). Each dataset provides 500 training samples per class. The number of test samples per class is 100 for Split CIFAR-100 and Mini-ImageNet, and 50 for Split Tiny-ImageNet.

**In addition to the standard 10-task Split CIFAR-100 protocol, we evaluate longer task sequences by repartitioning CIFAR-100 into 20 tasks with 5 classes per task and 50 tasks with 2 classes per task. These settings keep the dataset and buffer size fixed while increasing the number of sequential learning stages, providing a more challenging test of scalability under longer CL horizons (see Section 5.3 and Appendix A.6).**

**Out-of-Distribution Evaluation.** To assess robustness under natural distribution shifts, we construct an OOD evaluation set by applying a subset of corruption types from the ImageNet-C benchmark Hendrycks & Dieterich (2019). Specifically, we apply 11 corruptions: Gaussian noise, shot noise, impulse noise, defocus blur, motion blur, zoom blur, snow, fog, elastic transform, pixelate, and JPEG compression (see Figure 4 in Appendix A.2). Each corruption is applied at a fixed severity level, and all are applied independently to the test set. This results in a test set that is 12× larger than the original, consisting of the clean images and 11 corrupted variants per example. We exclude the brightness and contrast corruptions due to direct overlap with training-time augmentations used in contrastive learning (via ColorJitter). In addition, we exclude frost and frosted glass blur to avoid over-representing spatially obstructive or blur-based corruptions. While these two corruptions introduce distinct visual effects—such as textured overlays (frost) and irregular spatial blur (frosted glass blur)—their perceptual impact overlaps partially with other selected corruptions, such as snow, fog, and defocus blur, which also reduce visibility or introduce high-frequency distortions. This selection ensures that the OOD evaluation covers a broad range of realistic distribution shifts while avoiding redundancy and overlap with training-time augmentations.

To quantify performance under these shifts, we compute the mean accuracy across all corruptions and original images. We define OOD generalization as the ability of a CL model, trained on clean, ordered tasks, to maintain high predictive accuracy when evaluated under distribution shift. These shifts manifest as test-time corruptions that preserve semantic labels but alter low-level appearance.

Let a model trained on a sequence of tasks $\{\mathcal{D}_1^{\text{train}}, \ldots, \mathcal{D}_T^{\text{train}}\}$, sampled from an in-distribution source $\mathbb{P}_{\text{train}}$. Let $\mathcal{D}_{\text{test}}^{\text{iid}} \sim \mathbb{P}_{\text{iid}}$ denote the clean test set. We define a family of corruption functions $\mathcal{Q} = \{q_1, \ldots, q_K\}$, each inducing a corrupted distribution $\mathbb{P}_{\text{ood}}^{(k)}$ by transforming inputs from the clean test set: $\mathbb{P}_{\text{ood}}^{(k)} = \{(x', y) \mid x' = q_k(x), (x, y) \sim \mathbb{P}_{\text{iid}}\}, k = 1, \ldots, K$. For each corruption, we compute the model's classification accuracy: $\text{Acc}^{(k)} = \mathbb{E}_{(x,y) \sim \mathbb{P}_{\text{ood}}^{(k)}} \left[ \mathbb{I}\{\arg\max_{c \in \mathcal{C}_{1:t}} w_c^\top f_\theta(x) = y\} \right]$. We include the clean test accuracy $\text{Acc}^{(0)}$ and define the overall OOD generalization as the mean accuracy across clean and corrupted variants: $\text{Acc}^{\text{ood}} = \frac{1}{K+1} \sum_{k=0}^K \text{Acc}^{(k)}$. High OOD generalization indicates that the model's performance is stable under covariate shift, reflecting robustness to unseen visual conditions that differ from the training distribution.

**Metrics.** We evaluate CL performance using two standard metrics: Average Accuracy (ACC) and Backward Transfer (BWT) Lopez-Paz & Ranzato (2017). ACC quantifies overall performance by averaging the test accuracy across all tasks after the final training stage. BWT measures forgetting by computing the average performance drop on past tasks once the model has completed training. Formally, these metrics are defined as, $\text{ACC} = \frac{1}{T} \sum_{j=1}^T \alpha_{T,j}$, $\text{BWT} = \frac{1}{T-1} \sum_{j=1}^{T-1} \beta_{T,j}$. Where $\alpha_{i,j}$ denotes the accuracy on the test set of task $j$ after training up to task $i$, and $\beta_{i,j} = \alpha_{i,j} - \alpha_{j,j}$ captures the degradation in task $j$'s performance after subsequent training. $T$ denotes the total number of tasks. All metrics are reported under the class-incremental setting, following prior works Boschini et al. (2022); Tong et al. (2025).

**Baselines.** Our evaluation includes eleven rehearsal-based CL methods: GEM Lopez-Paz & Ranzato (2017), A-GEM Chaudhry et al. (2018b), ER Chaudhry et al. (2019), GSS Aljundi et al. (2019), GDUMB

Prabhu et al. (2020), MetaSP Sun et al. (2022), SOIF Sun et al. (2023), PCR Lin et al. (2023), ER-LAS Huang et al. (2023), DELTA Raghavan et al. (2024), and CSReL Tong et al. (2025).

**Implementation Details.** We adopt the implementations and hyperparameter configurations provided by the Mammoth framework Buzzega et al. (2020); Boschini et al. (2022), as well as those from MetaSP Sun et al. (2022), SOIF Sun et al. (2023), PCR Lin et al. (2023), ER-LAS Huang et al. (2023), DELTA Raghavan et al. (2024), and CSReL Tong et al. (2025). Additional details are included in Appendix A.3. All experiments employ ResNet-18 He et al. (2016) as the backbone architecture, trained from scratch. To ensure reliable comparison, each result is reported as the mean over five runs with fixed random seeds (0–4). We use a batch size of 32 for both current and memory samples. Training is conducted for 50 epochs per task using SGD, with learning rates of 0.1 on Split CIFAR-100 and 0.03 on both Split Mini-ImageNet and Split Tiny-ImageNet. Standard data augmentations, including random cropping and random horizontal flipping, are applied during training. For the reweighted softmax strategy, ~~we configure~~ the hyperparameters $\gamma_1$, $\gamma_2$, and $\gamma_3$ **are treated as global hyperparameters controlling the strength of the three correction factors in Eq. 11. They are not class-specific parameters and are not optimized separately for each** $\alpha_c$. ~~empirically~~ **Empirically** by default ~~as follows;~~ **, we use the following values:** ~~For~~ **for** Ours$_\beta$, Ours$_\gamma$, and Ours$_\lambda$, all $\gamma_1$, $\gamma_2$, and $\gamma_3$ values are set to 1.0 on Split CIFAR-100 and Split Mini-ImageNet, while on Split Tiny-ImageNet we use $\gamma_1 = 1.0$ and $\gamma_2 = \gamma_3 = 0.5$. For AA-RR, we set $\gamma_1 = 1.0$ and $\gamma_2 = \gamma_3 = 0.5$ for Split CIFAR-100 and Split Mini-ImageNet, and $\gamma_1 = 1.0$, $\gamma_2 = 0.5$, $\gamma_3 = 0$ for Split Tiny-ImageNet. Sensitivity analyses for these hyperparameters are reported in Section 5.3 and visualized in Figure 2. In our sample selection strategy, the hyperparameter $E$ is empirically set to 9 by default for both Ours$_\alpha$ and AA-RR. A sensitivity analysis of this setting is also provided in the ablation study (see Section 5.3, Figure 3).

## 5.2 Main Results

Table 1 reports ACC and BWT under both i.i.d. and OOD settings, comparing our approach against competitive baselines. To fairly isolate the effect of each component of our method, we evaluate four variants: Ours$_\alpha$, Ours$_\beta$, Ours$_\gamma$, and Ours$_\lambda$. Specifically, Ours$_\alpha$ integrates our sample selection strategy (see Section 4.3) into ER, using the standard softmax cross-entropy loss as in ER, GSS, MetaSP, SOIF, and CSReL. All these baselines, except ER, incorporate buffer update strategies, enabling a fair comparison. As shown, our selection strategy consistently outperforms these methods on average.

In Ours$_\beta$, Ours$_\gamma$, and Ours$_\lambda$, we replace the reweighting mechanisms of PCR, DELTA, and ER-LAS, respectively, with our own reweighting strategy (see Section 4.1), while keeping reservoir sampling. Note that PCR and DELTA use contrastive learning and therefore operate on both original and augmented samples. This setup improves their OOD performance but often degrades i.i.d. performance compared to ER-LAS. Our approach achieves stronger average performance across both settings compared to these methods.

Finally, AA-RR integrates all components—sample selection, reweighting, and contrastive-style augmentation (see Section 4.2). It achieves the highest ACC in OOD settings**, which is the primary focus of this work**, **while** remaining competitive under i.i.d. conditions. **We also observe that several AA-RR components improve standard i.i.d. performance: for example, Ours$_\alpha$ improves over buffer-selection baselines, and Ours$_\lambda$ achieves strong i.i.d. accuracy when our reweighting strategy is integrated with ER. These results indicate that the proposed mechanisms are not only beneficial under covariate shift, but can also improve standard CL retention.** Regarding BWT, some baselines report favorable values in OOD settings, but these often coincide with low ACC, suggesting that the models failed to learn sufficient knowledge to retain, much less forget.

**Comparison with Traditional OOD Generalization Methods.** To address the relationship between AA-RR and the stationary OOD literature discussed in Section 3.5 empirically, we re-implement four representative OOD-inspired strategies inside the same rehearsal framework used by ER and compare them with AA-RR on Split CIFAR-100 with buffer size 1000. The baselines are: ER-Aug—ER with the same SupCon-style augmentation pipeline (Khosla et al., 2020) that AA-RR uses, applied to both current and buffer batches; ER-Fact—ER with SupCon-style augmentation pipeline (same as AA-RR) and a Barlow-Twins–style factor-

Table 1: Results for class-incremental learning: Average Accuracy (ACC; higher is better) and Backward Transfer (BWT; less negative is better) under i.i.d. and OOD settings, averaged over 5 runs. The 'Mean' column shows the average performance of each method across all datasets. Best results are bolded, and second-best are underlined. Note: BWT for GDUMB is omitted due to computational constraints.

| Method | Dataset | Split CIFAR-100 500 i.i.d. | OOD | 1000 i.i.d. | OOD | 2000 i.i.d. | OOD | Split Mini-ImageNet 500 i.i.d. | OOD | 1000 i.i.d. | OOD | 2000 i.i.d. | OOD | Split Tiny-ImageNet 500 i.i.d. | OOD | 1000 i.i.d. | OOD | 2000 i.i.d. | OOD | Mean i.i.d. | OOD |
|---|---|---|---|---|---|---|---|---|---|---|---|---|---|---|---|---|---|---|---|---|---|
| GEM | ACC | 21.33 | 3.36 | 31.91 | 4.69 | 36.33 | 4.98 | 18.04 | 2.97 | 19.02 | 2.97 | 18.99 | 2.94 | 6.07 | 1.18 | 6.92 | 1.20 | 7.08 | 1.22 | 18.41 | 2.83 |
| | BWT | -66.83 | -18.85 | -48.93 | -15.75 | -34.40 | -12.81 | -51.53 | -9.09 | -46.69 | -8.33 | -42.43 | -7.12 | -46.84 | -8.75 | -41.11 | -7.80 | -35.97 | -6.59 | -46.08 | -10.57 |
| A-GEM | ACC | 9.32 | 2.31 | 9.23 | 2.39 | 9.25 | 2.36 | 14.65 | 2.72 | 14.69 | 2.82 | 14.74 | 2.86 | 6.30 | 1.30 | 6.30 | 1.33 | 6.23 | 1.29 | 10.08 | 2.15 |
| | BWT | -85.68 | -22.42 | -85.05 | -22.16 | -86.07 | -22.39 | -66.56 | -12.56 | -66.79 | -12.54 | -66.60 | -12.46 | -64.18 | -12.66 | -64.71 | -12.81 | -64.61 | -12.75 | -72.25 | -15.86 |
| GDUMB | ACC | 10.36 | 1.76 | 16.35 | 2.45 | 25.47 | 3.33 | 7.09 | 1.49 | 10.15 | 1.83 | 14.33 | 2.33 | 3.59 | 0.79 | 5.15 | 0.93 | 7.53 | 1.17 | 11.11 | 1.78 |
| | BWT | - | - | - | - | - | - | - | - | - | - | - | - | - | - | - | - | - | - | — | — |
| ER | ACC | 19.29 | 3.32 | 24.46 | 3.87 | 31.94 | 4.53 | 18.02 | 3.16 | 20.78 | 3.39 | 24.72 | 3.67 | 8.90 | 1.44 | 10.75 | 1.61 | 13.88 | 1.86 | 19.19 | 2.98 |
| | BWT | -75.53 | -19.69 | -69.29 | -18.80 | -61.18 | -17.79 | -63.62 | -11.60 | -60.07 | -10.98 | -55.12 | -9.68 | -68.53 | -12.40 | -67.12 | -12.05 | -63.47 | -11.21 | -64.88 | -13.80 |
| GSS | ACC | 22.23 | 3.45 | 28.84 | 4.13 | 35.27 | 4.87 | 18.48 | 3.19 | 21.91 | 3.43 | 28.16 | 3.96 | 9.23 | 1.49 | 11.45 | 1.70 | 15.52 | 2.03 | 21.23 | 3.14 |
| | BWT | -70.26 | -16.85 | -61.81 | -15.86 | -53.51 | -13.79 | -61.72 | -11.43 | -55.95 | -10.57 | -46.80 | -8.80 | -67.56 | -11.64 | -64.05 | -11.00 | -58.06 | -9.74 | -59.97 | -12.19 |
| MetaSP | ACC | 21.87 | 3.78 | 26.23 | 3.94 | 29.74 | 4.49 | 22.92 | 3.57 | 27.84 | 4.03 | 33.39 | 4.61 | 11.34 | 1.70 | 14.72 | 1.95 | 19.22 | 2.36 | 23.03 | 3.38 |
| | BWT | -71.01 | -16.03 | -65.53 | -16.04 | -59.90 | -14.01 | -60.40 | -10.78 | -53.28 | -9.44 | -44.64 | -8.04 | -67.63 | -11.73 | -63.03 | -10.97 | -56.61 | -9.73 | -60.23 | -11.86 |
| SOIF | ACC | 18.85 | 3.18 | 26.19 | 3.96 | 35.36 | 4.03 | 18.75 | 3.23 | 22.85 | 3.56 | 23.46 | 3.68 | 9.10 | 1.46 | 11.66 | 1.69 | 11.58 | 1.65 | 18.64 | 2.94 |
| | BWT | -71.47 | -19.72 | -62.91 | -17.04 | -62.11 | -16.44 | -65.70 | -12.35 | -58.65 | -11.39 | -54.76 | -10.74 | -69.63 | -12.61 | -65.65 | -12.04 | -63.64 | -11.19 | -63.84 | -13.72 |
| CSReL | ACC | 26.61 | 3.95 | 32.27 | 4.51 | 35.18 | 4.89 | 23.31 | 3.84 | 28.25 | 4.23 | 34.54 | 4.67 | 11.50 | 1.63 | 15.57 | 2.00 | 21.15 | 2.52 | 25.38 | 3.58 |
| | BWT | -61.31 | -17.18 | -51.37 | -14.45 | -44.39 | -13.53 | -57.60 | -10.84 | -48.51 | -8.93 | -36.68 | -7.27 | -65.44 | -11.89 | -60.05 | -10.76 | -50.83 | -9.16 | -52.91 | -11.56 |
| Ours$_\alpha$ | ACC | 28.77 | 4.36 | 34.91 | 4.95 | 42.18 | 5.57 | 24.14 | 3.66 | 30.58 | 4.35 | 36.50 | 4.78 | 11.58 | 1.73 | 16.54 | 2.18 | 23.27 | 2.79 | 27.61 | 3.81 |
| | BWT | -59.56 | -16.31 | -50.55 | -13.98 | -39.99 | -12.54 | -56.66 | -10.06 | -47.15 | -8.47 | -36.89 | -6.94 | -66.03 | -11.72 | -59.02 | -10.28 | -49.56 | -8.73 | -51.71 | -11.00 |
| PCR | ACC | 23.34 | 11.61 | 33.03 | 15.01 | 37.63 | 17.03 | 17.08 | 8.28 | 20.20 | 9.01 | 24.77 | 10.62 | 10.05 | 4.57 | 13.36 | 5.42 | 16.37 | 6.14 | 21.76 | 9.74 |
| | BWT | -58.22 | -34.36 | -48.70 | -31.34 | -49.98 | -35.31 | -61.13 | -30.38 | -57.65 | -30.92 | -52.41 | -29.85 | -60.48 | -30.56 | -55.96 | -29.99 | -51.24 | -29.24 | -55.09 | -31.33 |
| Ours$_\beta$ | ACC | 28.65 | 13.43 | 34.95 | 15.92 | 40.74 | 18.15 | 23.17 | 10.26 | 26.02 | 11.09 | 29.90 | 12.59 | 10.55 | 4.65 | 12.87 | 5.09 | 16.75 | 6.28 | 24.85 | 10.83 |
| | BWT | -47.60 | -21.31 | -42.18 | -19.51 | -40.06 | -20.16 | -46.33 | -19.80 | -46.09 | -21.64 | -42.62 | -21.75 | -56.87 | -26.53 | -52.15 | -24.64 | -50.38 | -26.31 | -47.14 | -22.41 |
| DELTA | ACC | 13.92 | 6.74 | 19.88 | 8.39 | 28.25 | 11.71 | 16.41 | 6.77 | 19.07 | 7.18 | 24.36 | 8.87 | 8.03 | 2.99 | 9.21 | 3.33 | 12.99 | 4.05 | 16.90 | 6.67 |
| | BWT | -79.24 | -42.02 | -71.41 | -38.70 | -59.83 | -31.11 | -61.32 | -24.59 | -56.77 | -23.04 | -48.48 | -19.86 | -64.88 | -26.55 | -62.44 | -25.18 | -55.95 | -21.85 | -62.26 | -28.10 |
| Ours$_\gamma$ | ACC | 21.04 | 8.08 | 28.08 | 10.60 | 36.88 | 13.71 | 22.53 | 7.34 | 25.56 | 8.27 | 29.39 | 9.55 | 10.87 | 3.35 | 12.69 | 3.68 | 15.44 | 4.40 | 22.50 | 7.67 |
| | BWT | -56.33 | -22.62 | -48.56 | -19.91 | -38.26 | -15.64 | -39.63 | -13.65 | -35.05 | -12.22 | -29.25 | -10.00 | -54.16 | -19.76 | -49.09 | -17.20 | -41.63 | -14.14 | -43.55 | -16.13 |
| ER-LAS | ACC | 30.61 | 4.51 | 37.61 | 5.21 | 45.69 | 5.99 | 23.16 | 3.56 | 28.80 | 4.04 | 34.73 | 4.64 | 12.50 | 1.77 | 17.16 | 2.16 | 23.02 | 2.69 | 28.14 | 3.84 |
| | BWT | -59.76 | -14.84 | -49.48 | -12.09 | -37.88 | -10.11 | -55.61 | -9.71 | -46.01 | -8.25 | -37.82 | -6.44 | -62.06 | -11.02 | -55.03 | -9.24 | -45.77 | -7.67 | -49.94 | -9.93 |
| Ours$_\lambda$ | ACC | 36.56 | 5.18 | 43.60 | 5.81 | 48.67 | 6.51 | 28.41 | 3.90 | 33.14 | 4.27 | 37.42 | 4.64 | 16.37 | 2.05 | 20.23 | 2.38 | 24.51 | 2.84 | 32.10 | 4.18 |
| | BWT | -37.02 | -6.73 | -29.13 | -5.82 | -21.76 | -4.72 | -37.17 | -6.36 | -29.64 | -5.36 | -23.03 | -4.35 | -41.23 | -5.94 | -33.99 | -4.82 | -27.15 | -3.72 | -31.13 | -5.31 |
| AA-RR | ACC | 33.91 | 15.79 | 38.16 | 17.97 | 41.34 | 19.93 | 29.03 | 11.98 | 30.69 | 13.07 | 34.26 | 14.77 | 12.77 | 5.41 | 16.96 | 6.87 | 19.97 | 8.27 | 28.57 | 12.67 |
| | BWT | -27.89 | -8.35 | -24.11 | -5.48 | -18.15 | -2.67 | -33.56 | -12.87 | -34.60 | -12.39 | -29.42 | -10.50 | -50.38 | -21.04 | -42.28 | -17.13 | -37.80 | -15.07 | -33.13 | -11.72 |

ization regularizer (Zbontar et al., 2021) on the cross-correlation of the two views' features; ER-Mixup—ER with Mixup (Zhang et al., 2017) on the combined current and buffer batches with soft-label cross-entropy; and ER-GroupDRO—ER with task-level GroupDRO (Sagawa et al., 2019), maintaining an online log-weight per task and aggregating per-task losses through a softmax. All four methods use reservoir sampling, like ER. Since standard class-incremental CL does not provide domain labels, ER-GroupDRO uses task IDs (recovered from class labels via the disjoint per-task class sets) as groups and should be read as task-group robust weighting rather than domain-level GroupDRO. Implementation details and hyperparameter settings for these four OOD-inspired ER variants are provided in Appendix A.3.

Reading the design space of the methods through three axes—augmentation strategy, reweighting strategy, and buffer policy—makes the comparison precise. The following Table reports these axes alongside the ACC and BWT metrics under both i.i.d. and OOD evaluation. The augmentation, reweighting, and buffer signature of each method is shown explicitly. For example, AA-RR combines SupCon-style augmentation, the temporally adaptive AA-RR prior, and correctness-guided buffer management. Ours$_\beta$ drops the correctness-guided buffer and replaces it with reservoir sampling, while Ours$_\lambda$ further drops the augmentation. PCR and Ours$_\beta$ share the same augmentation (SupCon-style augmentation) but differ in reweighting (sample frequency in batch vs. AA-RR's CL-specific prior).

Three observations follow. First, strong augmentation alone is responsible for a sizeable share of the OOD ACC improvement. ER-Aug raises OOD ACC from 3.87 for ER to 10.13. However, this OOD ACC gain comes with greatly amplified OOD forgetting: ER-Aug's OOD BWT is $-48.40$, compared with $-18.80$ for ER. This suggests that much of the OOD improvement from augmentation alone comes at the cost of poor past-task retention. This isolates the role of

Comparison with traditional OOD generalization methods on Split CIFAR-100 with buffer size 1000. Each method is decomposed by augmentation, reweighting, and buffer policy. ACC and BWT are reported under both i.i.d. and OOD evaluation, averaged over 5 seeds. ACC is higher better; BWT is less negative better.

| Method | Augmentation | Reweighting | Buffer | ACC (i.i.d.) | ACC (OOD) | BWT (i.i.d.) | BWT (OOD) |
|---|---|---|---|---|---|---|---|
| ER | – | – | Reservoir | 24.46 | 3.87 | -69.29 | -18.80 |
| ER-Aug | SupCon | – | Reservoir | 24.22 | 10.13 | -67.45 | -48.40 |
| ER-Fact | SupCon + factorization | – | Reservoir | 22.49 | 10.04 | -70.28 | -49.33 |
| ER-Mixup | Mixup | – | Reservoir | 28.78 | 4.53 | -61.91 | -13.59 |
| ER-GroupDRO | – | GroupDRO (task) | Reservoir | 26.87 | 4.09 | -65.39 | -16.34 |
| ER-LAS | – | Frequency prior | Reservoir | 37.61 | 5.21 | -49.48 | -12.09 |
| Ours$_\lambda$ | – | AA-RR prior | Reservoir | **43.60** | 5.81 | -29.13 | -5.82 |
| PCR | SupCon | Freq.-contrastive | Reservoir | 33.03 | 15.01 | -48.70 | -31.34 |
| Ours$_\beta$ | SupCon | AA-RR prior | Reservoir | 34.95 | 15.92 | -42.18 | -19.51 |
| AA-RR | SupCon | AA-RR prior | Correctness | 38.16 | **17.97** | **-24.11** | **-5.48** |

the augmentation component in Section 4.2—it is central to substantial OOD ACC—but also shows that augmentation alone cannot deliver the paper's goal, which is OOD performance that persists across tasks. Adding a Barlow-Twins–style factorization regularizer on top of strong augmentation, as in ER-Fact, does not change this pattern: ER-Fact obtains OOD ACC of 10.04 and OOD BWT of $-49.33$, while slightly reducing i.i.d. ACC.

Second, the AA-RR adaptive prior, when combined with augmentation, restores OOD BWT and pushes OOD ACC further. Going from ER-Aug to Ours$_\beta$, which adds the AA-RR prior on top of the same augmentation while retaining reservoir sampling, moves BWT from ($-67.45$, $-48.40$) to ($-42.18$, $-19.51$) for i.i.d. and OOD settings, respectively, and lifts OOD ACC from 10.13 to 15.92. The asymmetric drop in OOD BWT that augmentation introduces is consistent with the gradient-imbalance analysis in Section 3.2: strong augmentation doubles the number of training views for both current and replay samples per batch, but only $N_b$ unique exemplars underlie the buffer-side augmented views—far fewer than the $N_t$ unique samples available on the current-task side—so the augmented buffer cannot fully match the representational diversity entering the encoder from current-task data; this can aggravate current-task dominance in encoder updates and may worsen the feature drift for past classes identified in Section 3.2. The adaptive prior helps counteract this effect by redistributing the effective per-class gradient signal back toward past classes (Section 4.1). PCR, which also uses SupCon-style augmentation but relies on the sample frequency within batch, achieves slightly lower OOD ACC than Ours$_\beta$, 15.01 versus 15.92, and substantially worse OOD BWT, $-31.34$ versus $-19.51$—consistent with the prior-adjustment view in Section 4.1, where the CL-specific prior tracks task progression and class age while frequency-based priors do not.

Third, task-level GroupDRO and Mixup provide limited OOD benefit in this setting. ER-GroupDRO and ER-Mixup obtain OOD ACC values close to ER, 4.09 and 4.53, respectively, versus 3.87, while improving i.i.d. ACC modestly (26.87 and 28.78), and improving both i.i.d. and OOD BWT modestly. We attribute this to the structure of class-incremental CL: tasks are accessed in a strict order, so during training of task $t$, only task $t$ contributes fresh gradient, while previous tasks are represented only through the limited buffer—softmax-weighted reweighting of per-task losses does not, on its own, sufficiently counteract the recency- and dilution-driven imbalance characterized in Section 3.2. Likewise, mixing samples across the current/buffer split does not directly address the asymmetric gradient polarity that drives feature drift.

**Final class/task distribution in the buffer.** We analyze the buffer at the end of training for each method, extracting the number of samples per class and task. Subsequently, we calculate the Coefficient of Variation (CV) for classes and tasks using the formula $\left(\frac{\text{Standard Deviation}}{\text{Mean}}\right) \times 100\%$. The results are presented in Table 2. A CV of 0.00 signifies perfect balance, with higher values indicating greater imbalance.

Only our buffer update policy (used in Ours$_\alpha$ and AA-RR) achieves perfectly class- and task-balanced buffer distributions, with a CV of 0.00.

Table 2: Coefficient of Variation (CV) of class/task distribution in the buffer at the end of training on Split CIFAR-100 (buffer: 1000).

| Method | GEM | A-GEM | ER | GSS | MetaSP | SOIF | CSReL | Ours$_\alpha$ | PCR | Ours$_\beta$ | DELTA | Ours$_\gamma$ | ER-LAS | Ours$_\lambda$ | AA-RR |
|---|---|---|---|---|---|---|---|---|---|---|---|---|---|---|---|
| CV of Tasks | 0.00 | 0.00 | 06.80 | 58.79 | 26.96 | 29.87 | 0.00 | 0.00 | 06.80 | 06.80 | 06.80 | 06.80 | 06.80 | 06.80 | 0.00 |
| CV of Classes | 28.91 | 29.77 | 31.43 | 68.38 | 53.85 | 63.97 | 44.25 | 0.00 | 30.56 | 28.98 | 33.59 | 31.75 | 31.43 | 29.70 | 0.00 |

**Running Time.** Table 3 Compares the per-hour running time of each method on Split CIFAR-100 (buffer size: 2000), measured on a Quadro RTX 8000 GPU. Integrating our modules adds only negligible overhead on the baselines. Notably, methods with buffer update policies tend to incur high computational costs. Interestingly, AA-RR is even faster than PCR, as it updates the buffer only once per task—after training—rather than batch-by-batch as in reservoir sampling. Consequently, the buffer remains empty during the first task, further reducing AA-RR's runtime.

Table 3: Running times per hour for different methods on Split CIFAR-100 (buffer size 2000) using a Quadro RTX 8000.

| Method | GEM | A-GEM | ER | GSS | MetaSP | SOIF | CSReL | Ours$_\alpha$ | PCR | Ours$_\beta$ | DELTA | Ours$_\gamma$ | ER-LAS | Ours$_\lambda$ | AA-RR |
|---|---|---|---|---|---|---|---|---|---|---|---|---|---|---|---|
| Runtime (hours) | 20.42 | 1.85 | 1.4 | 15.56 | 2.87 | 8.13 | 2.52 | 1.4 | 3.6 | 3.68 | 3.78 | 3.98 | 1.42 | 1.5 | 3.55 |

## 5.3 Ablation.

**Correctness-Based Update Policies.** We ablate the buffer update policy in Ours$_\alpha$ by varying the selection criterion at epoch $E$. In addition to our default strategy of storing the most consistently correct samples ("Highest"), we evaluate three alternatives: "Lowest," "Middle," and "Uniform" sampling, all under class- and task-balance constraints.

As shown in Table 4, the "Highest" strategy consistently yields the best performance—except on Split CIFAR-100 (BWT, buffer size 2000), where it is nearly matched by "Middle." These results highlight the importance of informed sample selection for improving generalization and retention.

Table 4: Ablation of buffer update policy in Ours$_\alpha$. "Highest" selects the most consistently correct samples, "Middle" selects mid-correct samples, and "Lowest" selects the least correct.

| Setting | | | | | | | | | | |
|---|---|---|---|---|---|---|---|---|---|---|
| Dataset | Split CIFAR-100 | | | | Split Mini-ImageNet | | | | *Mean* | |
| Metric | ACC | | BWT | | ACC | | BWT | | ACC | BWT |
| Buffer | 1000 | 2000 | 1000 | 2000 | 1000 | 2000 | 1000 | 2000 | | |
| Uniform | 32.62 | 40.79 | -52.74 | -41.53 | 27.53 | 33.90 | -50.87 | -39.51 | 33.71 | -46.16 |
| Lowest | 20.45 | 26.80 | -66.53 | -58.67 | 19.70 | 23.65 | -60.59 | -53.15 | 22.65 | -59.74 |
| Middle | 33.99 | 41.58 | -51.55 | **-39.80** | 28.02 | 34.68 | -49.95 | -38.95 | 34.57 | -45.06 |
| Highest | **34.91** | **42.18** | **-50.55** | -39.99 | **30.58** | **36.50** | **-47.15** | **-36.89** | **36.04** | **-43.65** |

**Hyperparameter sensitivity.** We examine the sensitivity of the hyperparameters $\gamma_1$, $\gamma_2$, and $\gamma_3$ used in our softmax reweighting strategy of Ours$_\lambda$. In Figure 2, we fix $\gamma_1 = 1.0$ and vary $\gamma_2$ and $\gamma_3$ over $\{0.0, 0.5, 1.0, 1.5, 2.0\}$**, reporting both i.i.d. and OOD metrics (ACC and BWT)**. For experiments varying $\gamma_1 \in \{0.0, 0.5, 1.0, 1.5, 2.0\}$, see Appendix A.4. **The results show that performance is stable across a broad range of $\gamma_2$ and $\gamma_3$ values in both i.i.d. and OOD settings. Importantly, the OOD ACC and OOD BWT surfaces do not exhibit sharp degradation around the default setting, indicating that the robustness gains are not the result of a narrowly tuned hyperparameter configuration.**

In Figure 3, we examine the sensitivity of ~~our~~ **the** hyperparameter $E$, which ~~is integral to our memory~~ **controls the early-epoch window used by the correctness-guided buffer** update policy in Ours$_\alpha$. ~~As illustrated, the performance of hyperparameter $E$ remains stable across a wide range of epochs on all~~

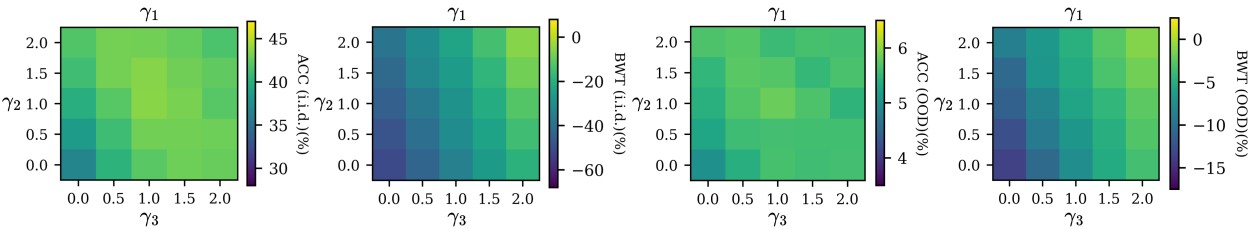

Figure 2: Sensitivity analysis of ~~our~~ **the** softmax reweighting hyperparameters in $Ours_\lambda$. **We fix $\gamma_1 = 1.0$ and vary $\gamma_2$ and $\gamma_3$ over $\{0.0, 0.5, 1.0, 1.5, 2.0\}$. We report ACC and BWT under both i.i.d. and OOD evaluation. For results varying $\gamma_1 \in \{0.0, 0.5, 1.0, 1.5, 2.0\}$, see Figure 5 in Appendix A.4. The results show that the performance for both i.i.d. and OOD generalization remains stable across a broad range of hyperparameter values, supporting the robustness of the proposed reweighting strategy.**

~~three datasets and buffer sizes, maintaining consistency in both ACC and BWT under i.i.d. conditions.~~ **We evaluate $E$ over a wide range of values and report ACC and BWT under both i.i.d. and OOD settings across all three datasets and buffer sizes. The results show that performance remains stable across $E$, indicating that the buffer-selection strategy is not sensitive to the exact length of the correctness-tracking window. This stability holds not only for standard i.i.d. evaluation but also for OOD robustness under label-preserving corruptions.**

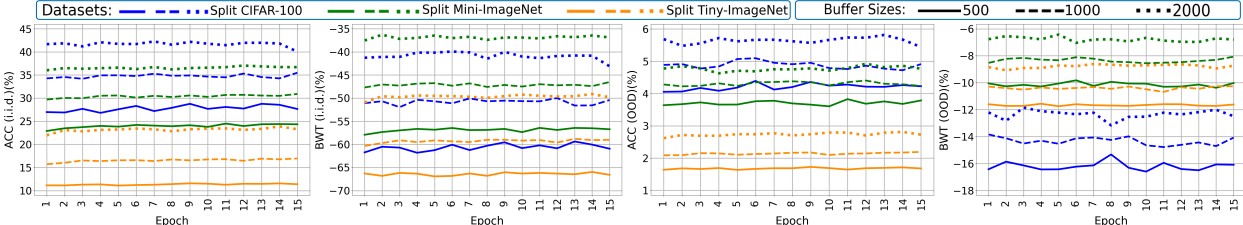

Figure 3: Sensitivity analysis of ~~hyperparameter~~ **the correctness-tracking window $E$** in $Ours_\alpha$. ~~Performance remains stable across a wide range of $E$ values, with consistent~~ **We report** ACC and BWT **under both i.i.d. and OOD evaluation** across Split CIFAR-100, Split Mini-ImageNet, and Split Tiny-ImageNet, ~~and~~ **with** buffer sizes 500, 1000, and 2000 ~~under i.i.d. conditions~~. **Performance remains stable across a wide range of $E$ values, indicating that both standard retention and OOD robustness are insensitive to the exact correctness-tracking window.**

**Performance on Split Mini-ImageNet (84×84 Resolution).** Appendix A.5 presents results on Split Mini-ImageNet using the original image resolution of 84×84 pixels and a buffer size of 1000, to assess the impact of image downscaling on performance.

**Scalability to Longer Task Sequences. Appendix A.6 presents results on Split CIFAR-100 (buffer size 1000) under two longer task sequences—20 tasks of 5 classes and 50 tasks of 2 classes—in addition to the 10-task, 10-class setting reported in Table 1. As expected, performance degrades as the number of tasks increases, reflecting the greater difficulty of CL over longer sequences. Nevertheless, AA-RR consistently achieves the best OOD ACC, i.i.d. BWT, and OOD BWT across both alternative task regimes. For i.i.d. ACC, $Ours_\lambda$ remains the top performer in both settings, mirroring the trend observed in Table 1. These results demonstrate that the OOD robustness advantage of AA-RR persists even under substantially longer**

**task sequences. Together with the memory-complexity analysis in Section 4.3 and the runtime results in Table 3, they suggest that AA-RR's lightweight design—requiring only three global reweighting hyperparameters and a per-task array of $N_t$ scalar counters—scales effectively to longer task horizons without sacrificing OOD robustness.**

## 6 Conclusion

In this work, we addressed a critical yet underexplored limitation of rehearsal-based continual learning (CL): ~~its vulnerability to overfitting and poor generalization under distributional shift~~ although standard (i.i.d.) generalization in CL has received attention in prior work, the robustness of rehearsal-based methods under label-preserving covariate shifts at test time has not been systematically examined, and we showed that current state-of-the-art methods degrade sharply in this regime. We introduced AA-RR, a lightweight framework that integrates structured adaptive class-aware reweighting, principled sample selection, and targeted data-centric augmentation ~~to mitigate the effects of gradient imbalance, spurious correlations, and buffer-induced class imbalance~~. The reweighting and selection components stabilize past-task representations under non-stationary training—a prerequisite for retaining any structure, OOD-robust or otherwise—while the augmentation component targets covariate-shift robustness directly. Through extensive experiments, we showed that AA-RR not only improves in-distribution performance but also significantly enhances robustness to covariate shifts, outperforming existing state-of-the-art methods. Our findings underscore the importance of designing CL algorithms that not only retain past knowledge but also preserve generalization beyond the training distribution.

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

## A Appendix

### A.1 Algorithm

The overall AA-RR training procedure is summarized in Algorithm 1, illustrating how our components are integrated into CL.

### A.2 OOD Images

Figure 4 illustrates the various corruption types applied to generate OOD samples from in-distribution data. These corruptions simulate distributional shifts that commonly occur in real-world settings, enabling a more robust evaluation of model generalization. The figure includes noise-based corruptions such as Gaussian noise, shot noise, and impulse noise; blur types including defocus, motion, and zoom blur; as well as appearance-altering transformations like fog, snow, elastic distortion, pixelation, and JPEG compression. Each corrupted version is derived from the same original image (top-left), demonstrating the visual impact of each corruption type used for OOD data.

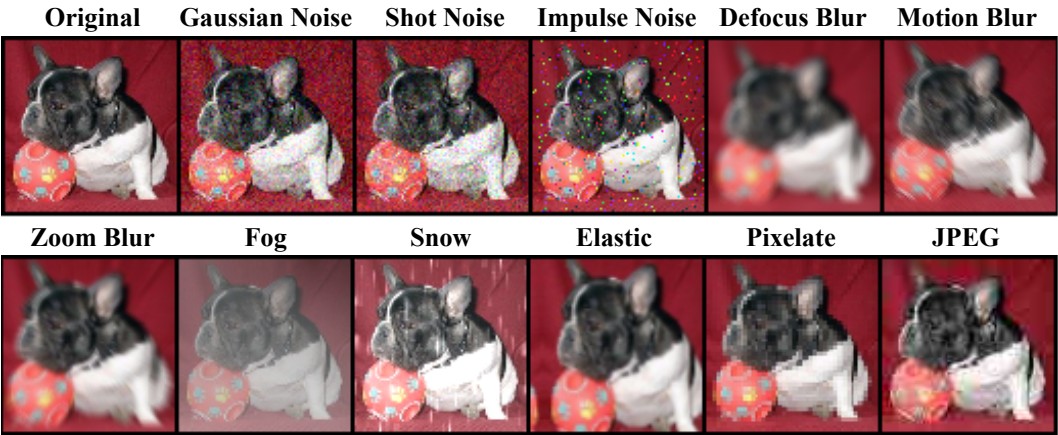

Figure 4: Examples of corruption types used to generate OOD samples. Starting from the original image (top-left), various corruptions such as Gaussian Noise, Shot Noise, Impulse Noise, different types of blur (Defocus, Motion, Zoom), and artifacts like Fog, Snow, Elastic distortions, Pixelation, and JPEG compression are applied to simulate distributional shifts.

---

**Algorithm 1** Adaptive Augmented Reweighted Replay (AA-RR)

---

**Require:** Task sequence $\{\mathcal{D}_1, \ldots, \mathcal{D}_T\}$, buffer size $B$, temperature $\tau$, **global** reweighting **hyper**parameters $\gamma_1, \gamma_2, \gamma_3$, early-epoch window $E$, task epochs $H$

1: Initialize model $\theta$, classifier weights $\{w_c\}$, buffer $\mathcal{B} \leftarrow \emptyset$
2: **for** task $t = 1$ to $T$ **do**
3:     **Initialize per-sample correctness counters** $r_i \leftarrow 0$ **for** $i = 1, \ldots, N_t$
4:     Compute per-class reweighting factor $\alpha_c$:

$$\alpha_c = \begin{cases} 1, & \text{if } c \in \mathcal{C}_t \\ \left(\dfrac{1}{t-1}\right)^{\gamma_1} \cdot \left(\dfrac{1}{t-t_c}\right)^{\gamma_2} \cdot \left(\dfrac{f_c}{\bar{f}_{\mathcal{C}_t} + \hat{n}}\right)^{\gamma_3}, & \text{if } c \in \mathcal{C}_{<t} \end{cases}$$

5:     **for** epoch $e = 1$ to $H$ **do**
6:         **for** batch $(x, y, x_{\text{orig}}, i)$ in $\mathcal{D}_t$ **do**
7:             **Current task:**
8:             Apply strong SupCon-style augmentation: $x' \leftarrow \text{Augment}(x)$
9:             Concatenate: $x_{\text{cur}} \leftarrow [x; x']$, $y_{\text{cur}} \leftarrow [y; y]$
10:            Get embedding: $z_{\text{cur}} \leftarrow f_\theta(x_{\text{cur}})$
11:            **if** $e \leq E$ **then**
12:                **Correctness tracking:**
13:                For each $(x_i^{ori}, y_i) \in \mathcal{D}_t$, ~~store~~ **update** correctness **counter**:

$$\cancel{c_{e,i}}\ r_i \leftarrow r_i + \mathbf{1}\left[\arg\max_{c \in \mathcal{C}_{1:t}} w_c^\top f_\theta(x_i^{ori}) = y_i\right]$$

14:            **end if**
15:            **Buffer:**
16:            **if** $\mathcal{B} \neq \emptyset$ **then**
17:                Sample $(x_b, y_b)$ from buffer
18:                Apply strong SupCon-style augmentation: $x_b' \leftarrow \text{Augment}(x_b)$
19:                Concatenate: $x_{\text{buf}} \leftarrow [x_b; x_b']$, $y_{\text{buf}} \leftarrow [y_b; y_b]$
20:                Get embedding: $z_{\text{buf}} \leftarrow f_\theta(x_{\text{buf}})$
21:                Combine: $z \leftarrow [z_{\text{cur}}; z_{\text{buf}}]$, $y \leftarrow [y_{\text{cur}}; y_{\text{buf}}]$
22:            **else**
23:                $z \leftarrow z_{\text{cur}}$, $y \leftarrow y_{\text{cur}}$
24:            **end if**
25:            Compute cosine-**similarity** logits: $\ell_c = \cancel{\dfrac{\tilde{w}_c^\top \tilde{z}}{\tau}}\ \text{sim}(\tilde{w}_c, \tilde{z})/\tau$
26:            Compute reweighted softmax cross-entropy loss:

$$\mathcal{L} = -\log\left(\frac{e^{\ell_y} \cdot \alpha_y}{\sum_{j \in \mathcal{C}_{1:t}} e^{\ell_j} \cdot \alpha_j}\right)$$

27:            Update model $\theta$, and classifier weights $\{w_c\}$ via gradient descent
28:         **end for**
29:     **end for**
30:     **Buffer Management:**
31:     ~~Compute total correctness scores: $r_i = \sum_{e=1}^{E} c_{e,i}$, for all $x_i^{ori} \in \mathcal{D}_t$~~
32:     Allocate per-task buffer budget: $m_t \leftarrow \lfloor B/t \rfloor$
33:     For each class $c \in \mathcal{C}_t$: $m_{t,c} \leftarrow \lfloor m_t/|\mathcal{C}_t| \rfloor$
34:     Select top-$m_{t,c}$ current-task samples by $r_i$
35:     For each prior class $c \in \mathcal{C}_{<t}$, retain **the first** ~~top-~~$m_{i,c}$ **samples** ~~from~~ **in the** existing **per-class** buffer **ordering**
36:     Update buffer: $\mathcal{B} \leftarrow \bigcup_{i=1}^{t} \bigcup_{c \in \mathcal{C}_i} \mathcal{S}_{i,c}$ where $|\mathcal{S}_{i,c}| = m_{i,c}$
37: **end for**
38: **Output:** Updated model parameters $\theta$, classifier weights $\{w_c\}$, and buffer $\mathcal{B}$

---

### A.3 Implementation Details

For SOIF, we follow the original paper and set the probability of selecting an already-in-coreset sample to 0.5, the weight for second-order influence functions to 0.01, and the NTK regularization coefficient to 1e-3. For PCR and DELTA, we use the temperature value of 0.09, consistent with their respective implementations. For ER-LAS, we adopt the default settings from the original paper, setting the logit adjustment parameter and the sliding window length both to 1. For CSReL hyperparameters, we used the following configurations. On Split CIFAR-100, the main continual learner was trained with a learning rate of 2e-2, a loss factor of 4.0, and 40 selection steps. The selected subset was trained with a learning rate of 5e-3 for 7 steps. The holdout model was trained for 10 epochs with a learning rate of 3e-3 and 200 samples per previous task. For Split Mini-ImageNet and Split Tiny-ImageNet, we maintained the main learner's learning rate at 2e-2 and the loss factor at 4.0, but increased the selection steps to 250. The selected subset was trained with a learning rate of 5e-3 for 20 steps. The holdout model used the same learning rate (3e-3), but was trained for 20 epochs using 200 samples per previous task.

**Traditional OOD Baselines.** For the four traditional OOD generalization baselines compared in Section 5.2, we use the following hyperparameters. ER-GroupDRO uses task-level Group-DRO (Sagawa et al., 2019): the group-weight step size is set to $\eta = 0.01$, following the default used by the WILDS benchmark (Koh et al., 2021) for the GroupDRO algorithm; groups are defined by task IDs, mapping labels to task indices via the disjoint per-task class sets. ER-Mixup applies Mixup (Zhang et al., 2017) to the combined current/replay batch: the Beta-distribution parameter is set to $\alpha = 1.0$, following the CIFAR-scale Mixup setting. ER-Fact adds a Barlow-Twins-style factorization regularizer (Zbontar et al., 2021) between original and augmented representations: the factorization-regularizer weight added to the cross-entropy loss is set to $\lambda_{\text{fact}} = 0.1$, chosen from preliminary runs over $\{2, 1.5, 1, 0.5, 0.1, 0.01\}$; the Barlow-Twins off-diagonal coefficient inside the regularizer is fixed at $0.005$, following the standard Barlow-Twins setting. ER-Aug uses the same SupCon-style augmentation pipeline described in Section 4.2 and introduces no additional method-specific hyperparameters. All the other settings and hyperparameters are the same as reported in Section 5.1.

### A.4 Hyperparameter Sensitivity

Figure 5 presents a joint sensitivity analysis of the hyperparameters $\gamma_1$, $\gamma_2$, and $\gamma_3$ in ~~our~~ Ours$_\lambda$ ~~model~~ under ~~the i.i.d.~~ both i.i.d. and OOD settings. **This analysis complements Figure 2 by varying all three reweighting hyperparameters.** We systematically vary each parameter over the range $\{0.0, 0.5, 1.0, 1.5, 2.0\}$ and report the corresponding changes in ACC and BWT **in each setting**. For each value of $\gamma_1$ (shown across columns), we visualize the impact of jointly varying $\gamma_2$ (vertical axis) and $\gamma_3$ (horizontal axis)**; the four rows of panels report, from top to bottom, ACC (i.i.d.), BWT (i.i.d.), ACC (OOD), and BWT (OOD)**. The results indicate that performance is generally robust to moderate changes in these hyperparameters **across all four metrics**, with consistent improvements in backward transfer for higher values **under both i.i.d. and OOD evaluation. This evidence supports our description of Eq. 11 as a structured, empirically motivated reweighting rule rather than a configuration that depends on a precise hyperparameter choice, and confirms that the same stability holds for the OOD generalization claim that motivates the paper.**

### A.5 Performance on Split Mini-ImageNet (84×84 Resolution)

Table 5 reports results on Split Mini-ImageNet using the original image resolution of 84×84 pixels with a buffer size of 1000. This evaluation allows us to examine the effect of image resolution on **both standard i.i.d.** performance **and OOD robustness**. We compare methods under both i.i.d. and OOD settings, reporting ACC and BWT. All hyperparameters match those used in the main experiments, except for Ours$_\gamma$, where we set $\gamma_1 = 1.0$, $\gamma_2 = 0.5$, and $\gamma_3 = 0.5$.

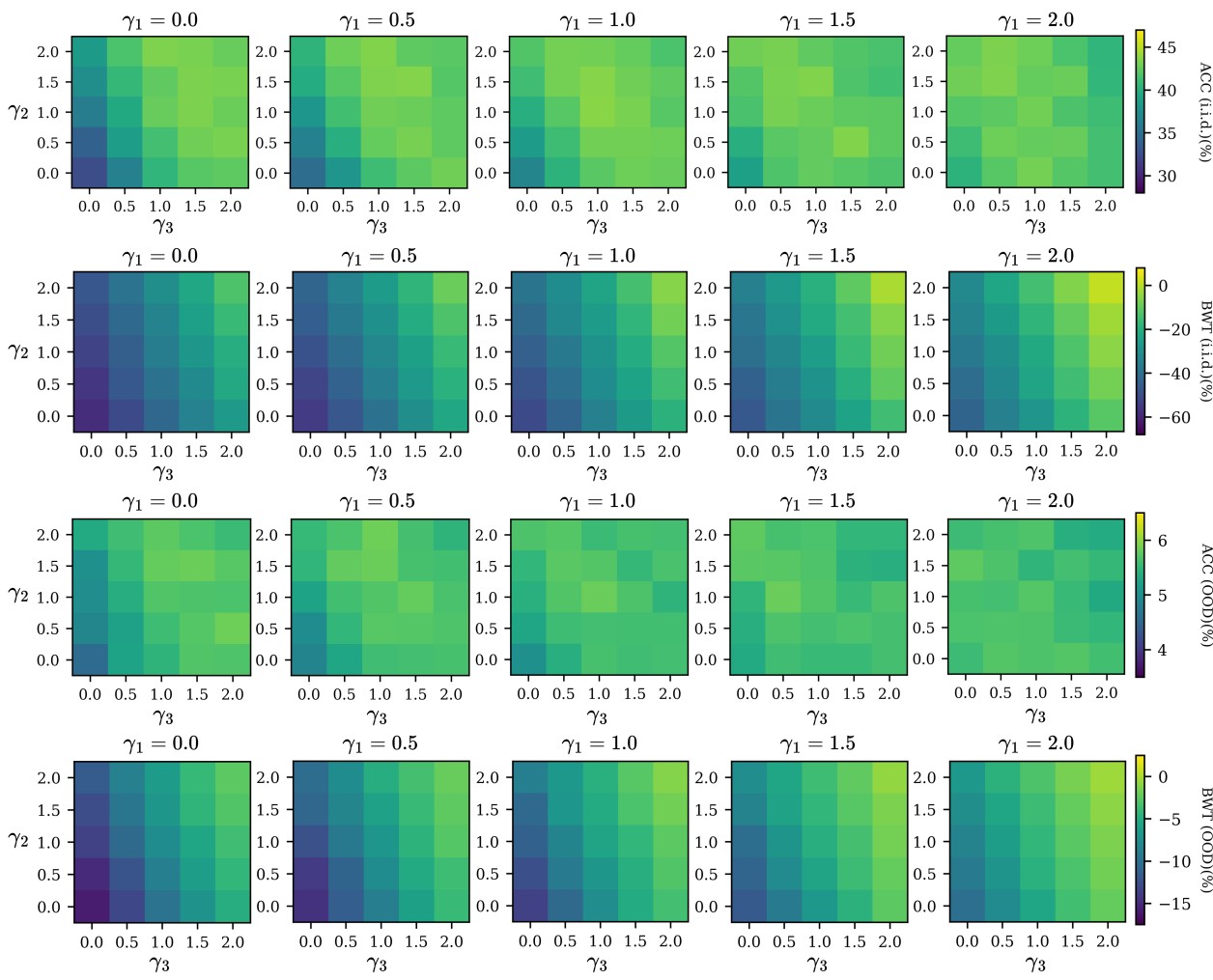

Figure 5: Joint sensitivity analysis of the hyperparameters $\gamma_1$, $\gamma_2$, and $\gamma_3$ in ~~our~~ Ours$_\lambda$ ~~model~~ under ~~the i.i.d. setting~~ **both i.i.d. and OOD evaluation**. Each parameter is systematically varied over the range $\{0.0, 0.5, 1.0, 1.5, 2.0\}$~~, and the resulting effects on ACC and BWT are reported~~. **The four rows of panels report, from top to bottom, ACC (i.i.d.), BWT (i.i.d.), ACC (OOD), and BWT (OOD).** For each value of $\gamma_1$ (arranged across columns), we visualize how changes in $\gamma_2$ (vertical axis) and $\gamma_3$ (horizontal axis) jointly affect performance. The results show that the model remains robust to moderate hyperparameter variation **across both standard and OOD evaluation**, with higher values generally leading to improved backward transfer **in both settings, supporting the use of Eq. 11 as a compact empirical reweighting rule rather than a heavily class-specific tuning procedure**.

## A.6 Scalability to Longer Task Sequences

To evaluate scalability under longer CL horizons, we additionally repartition Split CIFAR-100 into 20 tasks with 5 classes per task and 50 tasks with 2 classes per task, using a fixed buffer size of 1000. All other experimental settings follow the main experimental protocol in Section 5.1. By increasing the number of sequential tasks without changing the dataset or memory budget, these settings substantially increase the difficulty of replay-based CL.

Table 5: Results on Split Mini-ImageNet with the original image resolution of 84×84 pixels and a buffer size of 1000. Methods are compared under both i.i.d. and OOD settings, with ACC and BWT reported.

| Setting | Dataset / Method | GEM | A-GEM | GDUMB | ER | GSS | MetaSP | SOIF | CSReL | Ours$_\alpha$ | PCR | Ours$_\beta$ | DELTA | Ours$_\gamma$ | ER-LAS | Ours$_\lambda$ | AA-RR |
|---|---|---|---|---|---|---|---|---|---|---|---|---|---|---|---|---|---|
| i.i.d. | **ACC** | 30.34 | 16.48 | 13.71 | 26.57 | 24.34 | 35.09 | 29.14 | 35.09 | 36.37 | 27.53 | 30.25 | 24.47 | 30.09 | 36.61 | 44.25 | 36.00 |
| | **BWT** | -55.53 | -76.84 | - | -65.52 | -66.45 | -54.47 | -63.29 | -51.21 | -49.77 | -57.68 | -48.72 | -61.18 | -35.84 | -50.84 | -28.89 | -20.11 |
| OOD | **ACC** | 4.51 | 3.23 | 2.12 | 4.08 | 3.83 | 4.91 | 4.37 | 4.91 | 4.93 | 13.79 | 15.67 | 9.65 | 9.73 | 4.89 | 5.24 | 18.30 |
| | **BWT** | -10.34 | -13.43 | - | -10.52 | -11.26 | -9.10 | -11.47 | -9.01 | -8.21 | -38.10 | -27.82 | -28.16 | -14.23 | -7.82 | -4.54 | -7.75 |

**We consider a subset of methods: ER as the standard rehearsal baseline; CSReL as a strong policy-driven replay method; PCR and ER-LAS as two strong reweighting-based baselines; their corresponding AA-RR variants; and the full AA-RR method.**

Table 6: Results for class-incremental learning on Split CIFAR-100 under longer task sequences with buffer size 1000. We compare representative replay baselines and variants of our method under 20 tasks with 5 classes per task and 50 tasks with 2 classes per task. We report ACC and BWT under both i.i.d. and OOD evaluation, averaged over 5 seeds. The best result in each column is bolded. Higher ACC is better, and less negative BWT is better.

| Method | 20 tasks, 5 classes/task | | | | 50 tasks, 2 classes/task | | | |
|---|---|---|---|---|---|---|---|---|
| | ACC (i.i.d.) | ACC (OOD) | BWT (i.i.d.) | BWT (OOD) | ACC (i.i.d.) | ACC (OOD) | BWT (i.i.d.) | BWT (OOD) |
| ER | 19.85 | 3.04 | -72.70 | -27.59 | 13.04 | 2.06 | -80.63 | -46.31 |
| CSReL | 26.32 | 3.64 | -51.58 | -19.58 | 19.93 | 2.64 | -58.61 | -33.39 |
| Ours$_\alpha$ | 30.61 | 4.03 | -53.99 | -19.90 | 24.78 | 3.13 | -61.48 | -35.29 |
| PCR | 25.15 | 12.69 | -60.24 | -40.41 | 17.02 | 9.01 | -72.46 | -52.28 |
| Ours$_\beta$ | 28.35 | 12.90 | -34.52 | -14.07 | 21.28 | 9.62 | -16.75 | -6.19 |
| ER-LAS | 32.18 | 4.36 | -49.52 | -14.70 | 26.50 | 3.52 | -46.44 | -16.85 |
| Ours$_\lambda$ | **36.04** | 4.97 | -21.56 | -4.99 | **28.40** | 3.91 | -3.51 | -3.02 |
| AA-RR | 31.64 | **15.37** | **-7.45** | **3.81** | 24.36 | **11.37** | **5.66** | **6.27** |

**The results show that increasing the number of tasks substantially reduces ACC under both i.i.d. and OOD evaluation for all methods, confirming that these settings are more challenging than the standard 10-task protocol. Despite this increased difficulty, AA-RR maintains the strongest OOD performance in both settings. Notably, Ours$_\lambda$ achieves the highest i.i.d. ACC. The positive BWT values observed for AA-RR in the 50-task setting indicate that later training can improve performance on earlier tasks under this configuration.**

