# OpenReview forum: "Less Forgetting, More OOD Generalization: Adaptive Augmented Reweighted Replay (AA-RR) for Continual Learning"
_TMLR — Rejected by TMLR_

### Review · Reviewer_U6Jy · 2026-03-06

**Summary Of Contributions:**

This paper proposes continual learning with a memory buffer that addresses the OOD probelm, using a reweighted loss function to address forgetting of previous learning (reserved in the buffer, composed of selected samples from previous tasks). They consider the lag of first-class appearance in the weights and the mitigation of the overfitting problems. The authors examine experiments on a series of classification tasks with disjoint classes. The OOD classification accuracy and BWT (indicating the degree of performance degradation) were reported, and the results demonstrate the superiority of the proposed algorithm over the previous one. Furthermore, the ablation study concerning the components of the proposed algorithm.

**Additional Comments:**

None.

**Audience:**

Yes

**Audience Explanation:**

The topic is new. Basically, the continual learning is closely related to the OOD probelm. Therefore, the more solid research of this paper can be interesting.

**Broader Impact Concerns:**

None.

**Claims And Evidence:**

Yes

**Claims Explanation:**

The approaches and experiments can be convincing to ensure clear evidence.

1. The research on the OOD tasks in continual learning is an interesting topic, and the proposed algorithm is impressive in achieving performance on some image datasets.
2. The algorithm can tackle the various aspects of continuous learing with the buffer.

**Requested Changes:**

1. The re-weight can be viewed as the prior. The paper can be strengthened by explanations or research on these aspects.
2. The experiments with more intensive datasets or a large number of tasks can be valuable to make the paper stronger, which is not included in this paper.
3. It is better to describe the relationship between the traditional OOD algorithms (such as weighting and others) and the proposed algorithm. It is not easy; the experiments concerning this issue can be interesting.

---

> ### Author Response · Authors · 2026-05-06
>
> >1. The re-weight can be viewed as the prior. The paper can be strengthened by explanations or research on these aspects.
>
> We thank the reviewer for this insightful comment. We agree that the proposed reweighting factor can be interpreted from a prior-adjustment perspective, since it modifies the softmax normalization in a way that changes the effective class prior during training. This connection is indeed important and can help clarify the motivation of our method.
>
> In the revised manuscript, we will strengthen the explanation by explicitly discussing the relationship between our reweighting strategy and prior-based logit adjustment. In particular, existing methods such as LAS and DELTA adjust logits using estimated class priors derived from recent batches or cumulative class frequencies. However, these priors are primarily frequency-based and do not explicitly account for the temporal nature of continual learning, such as class recency, task age, and buffer dilution. Our weighting factor extends this prior-adjustment view by incorporating not only class frequency but also task progression and class staleness, making the effective prior adaptive to the continual learning process.
>
> More specifically, the proposed weight $\alpha_c$, Eq. 11, can be viewed as a temporally adaptive prior correction for class $c$. Unlike static or batch-local priors, $\alpha_c$ reflects three CL-specific factors: the growing imbalance between current-task and buffered samples, the elapsed time since a class was introduced, and the repeated exposure of buffered examples. This allows the model to regulate gradient propagation across old and new classes more effectively under limited memory. We will add this interpretation to the method section and further clarify how it differs from prior-based reweighting approaches such as PCR, DELTA, and LAS, which are already discussed in Sections 3.3–3.4, and 4.1 of the paper.
>
>
> >2. The experiments with more intensive datasets or a large number of tasks can be valuable to make the paper stronger, which is not included in this paper.
>
> We thank the reviewer for this valuable suggestion. We agree that evaluating AA-RR on more intensive datasets or longer task sequences would further strengthen the empirical evidence.
>
> In the current submission, we focused on widely used class-incremental continual learning benchmarks, including Split CIFAR-100, Split Mini-ImageNet, and Split Tiny-ImageNet, to enable controlled and fair comparison with a broad set of rehearsal-based baselines. These benchmarks already cover different dataset scales and task configurations, and our results show consistent improvements under both i.i.d. and OOD evaluation settings.
>
> That said, we agree with the reviewer that testing under more demanding settings, such as larger-scale datasets or a larger number of sequential tasks, is important for further validating scalability. Following the reviewer’s suggestion, we are currently conducting additional experiments in a more challenging continual learning setting. Once the results become available, we will share them in the response. We will also revise the discussion section to explicitly acknowledge this point and clarify the scalability of AA-RR, including its lightweight design and low computational overhead.

---

> ### Author Response · Authors · 2026-05-06
>
> >3. It is better to describe the relationship between the traditional OOD algorithms (such as weighting and others) and the proposed algorithm. It is not easy; the experiments concerning this issue can be interesting.
>
> We thank the reviewer for this insightful suggestion. We agree that the relationship between traditional OOD generalization methods and the proposed AA-RR should be clarified further.
>
> Traditional OOD generalization methods typically address distribution shift in stationary training settings, where the training set is fixed and all data are jointly available. Common strategies include importance/class reweighting, robust data augmentation, invariant representation learning, group-robust optimization, and data-centric sample selection. AA-RR is related to these ideas, but the key distinction is that our setting is continual learning: the training distribution changes over time, past data are only accessible through a limited replay buffer, and replayed samples may be seen repeatedly. Therefore, the OOD problem in our paper is not only caused by covariate shift at test time, but also by CL-specific training dynamics such as buffer dilution, class imbalance, task recency, and overfitting to stored exemplars.
>
> In this sense, AA-RR can be viewed as adapting several traditional OOD generalization principles to rehearsal-based continual learning. First, our adaptive reweighting is related to prior correction or importance-weighting methods, but differs from standard OOD weighting because it does not estimate a target-domain density ratio. Instead, it corrects the effective training prior induced by non-stationary task order, class age, and replay-buffer statistics. Second, our augmentation module is related to OOD robustness methods that encourage invariance to low-level appearance changes, but we apply it in a replay setting to reduce overfitting to both current-task samples and repeatedly replayed memory samples. Third, our correctness-guided buffer selection is related to data-centric sample-selection methods, but it is designed for a fixed-capacity memory buffer and explicitly enforces class- and task-balanced retention.
>
> Following the reviewer’s suggestion, we are currently running an additional experiment to better examine this connection under the same rehearsal-based CL protocol. Once the results become available, we will share them in the response. We will also add a dedicated discussion clarifying how AA-RR relates to traditional OOD generalization methods and why CL-specific factors make this setting different from standard stationary OOD learning.

---

### Review · Reviewer_oSrw · 2026-03-11

**Summary Of Contributions:**

This paper addresses OOD generalization in rehearsal-based continual learning (CL), arguing that existing methods overfit to memorized replay samples and generalize poorly under covariate shift. The authors propose Adaptive Augmented Reweighted Replay (AA-RR), a framework with three components - adaptive re-weighting, augmentations, correctness based buffers. Experiments are conducted on Split CIFAR-100, Split Mini-ImageNet, and Split Tiny-ImageNet under both i.i.d. and OOD (corruption-based) evaluation settings, comparing against eleven rehearsal-based baselines.

### Strengths
- Studying OOD robustness in the continual learning setting is a meaningful direction. The paper clearly demonstrates (Main Table 1) that existing rehearsal-based methods exhibit significant performance degradation under covariate shift.
- The reweighting formula in Eq. 11, which adjusts class weights based on task progression, recency, and buffer frequency, is a novel and interesting contribution. The fact that weights change dynamically at each iteration rather than being static is potentially the most impactful component.
- The paper evaluates against a comprehensive set of eleven baselines and provides informative ablations (Ours-alpha through Ours-lambda) that isolate the contribution of each component.
- The method adds negligible computational overhead (Table 3), and the buffer update is performed only once per task rather than per-batch.
- The proposed selection strategy achieves perfect class- and task-balance (CV = 0.00 in Table 2), which is a clear advantage over reservoir sampling-based approaches.

### Weaknesses

-  The paper's central claim is about improving OOD generalization, yet much of the theoretical analysis (Section 3, particularly Section 3.2) addresses standard generalization which is well-studied in traditional CL. The distinction between "generalization" and "OOD generalization" is not clearly maintained throughout the paper. For example, the statement "... this overlooked aspect motivates our study..." (Section 1) overstates the novelty, since generalization (though not OOD generalization specifically) has been considered in prior CL work (which the authors themselves cite). The authors should be more precise in delineating where standard generalization analysis ends and where OOD-specific insights begin. However, it should be noted that results also improved on iid generalization as well.
- *"Principled" reweighting is not well justified.* The authors describe their reweighting as "principled," but the functional form in Eq. 11 is largely heuristic. Three tunable hyperparameters ($\gamma_1$, $\gamma_2$, $\gamma_3$) are introduced with dataset-specific tuning (e.g., different values for CIFAR-100 vs. Tiny-ImageNet). It is unclear why this particular multiplicative decomposition is principled as opposed to other possible formulations (One can simply use bayesian optimization over $\alpha_c$ instead?). The paper would benefit from a more formal justification or at least a clear acknowledgment that the formulation is empirically motivated.
- *Sensitivity analysis does not cover OOD accuracies.* Given that the main contribution is about OOD generalization, the sensitivity analyses of hyperparameters (Figure 2) and (Figure 3) only report i.i.d. ACC and BWT. The sensitivity of OOD performance to these hyperparameters is essential to assess the robustness of the claims.
- The entire analysis in Section 3 concerns gradient imbalance and generalization in the standard sense. There is no formal argument connecting unbalanced gradient propagation specifically to OOD performance degradation. The OOD assumption is not needed for the theoretical development in this section, which weakens the paper's core claim.
- *Missing memory complexity analysis.* The "correctness-guided buffer management" (Section 4.3) requires tracking per-sample correctness over epochs for each task, which involves storing prediction outcomes for all training samples. The authors should provide explicit memory complexity comparisons against a naive training baseline to quantify this overhead.

**Additional Comments:**

NA

**Audience:**

Yes

**Audience Explanation:**

Yes. The intersection of continual learning and OOD robustness is a relevant and under-explored area. The experimental findings, particularly the large gap between i.i.d. and OOD performance of existing methods, would be of interest to researchers working on continual learning, robustness, and distribution shift.

**Claims And Evidence:**

No

**Claims Explanation:**

Partially Yes. The experimental results convincingly demonstrate that AA-RR improves OOD accuracy over baselines. However, the theoretical motivation (Section 3) supports claims about standard generalization rather than OOD generalization specifically. The claim that the reweighting is "principled" is not adequately supported. The absence of OOD metrics in sensitivity analyses weakens the evidence for the robustness of the approach under its stated goal.

**Requested Changes:**

Please see the summary above for a list of changes which would make the article better.

---

> ### Author Response · Authors · 2026-05-06
>
> >- The paper's central claim is about improving OOD generalization, yet much of the theoretical analysis (Section 3, particularly Section 3.2) addresses standard generalization …
>
> We thank the reviewer for this careful and important comment. We agree that the distinction between standard/i.i.d. generalization and OOD generalization should be made more precise throughout the paper.
>
> The reviewer is correct that the gradient-imbalance analysis in Section 3.2 builds on machinery developed in prior CL work for explaining forgetting and standard generalization. In our paper, however, this analysis serves a different purpose: it diagnoses why past-task representations — including any OOD-robust structure acquired earlier in training — fail to persist across the task sequence. The gradient-polarity argument shows that old-class weights are continually pushed away from evolving feature distributions, causing feature drift; this drift erases OOD-relevant structure together with i.i.d. structure. In this sense, Section 3.2 is not a standard-generalization analysis transplanted from prior work, but a foundation for arguing that OOD-robust knowledge, once acquired, requires explicit preservation across tasks. We agree that this connection is currently implicit, and we will state it explicitly in the revision.
>
> Moreover, our intention was not to claim that generalization in continual learning has been overlooked in general. As the reviewer correctly points out, standard generalization and overfitting in CL have been studied in prior work, and our paper also cites this literature. The more specific gap we aim to address is OOD generalization under rehearsal-based continual learning, where models trained on clean sequential tasks are evaluated under label-preserving covariate shifts. We agree that some wording in the current manuscript, such as “this overlooked aspect motivates our study,” is too broad. We will revise such statements to make clear that the underexplored aspect is OOD generalization in rehearsal-based CL, rather than generalization in CL more generally.
>
> Finally, we will update the framing to reflect that AA-RR improves both i.i.d. and OOD performance, while the main contribution is identifying and improving the OOD robustness of rehearsal-based CL methods. This revision will make clear where standard CL generalization analysis ends and where the OOD-specific motivation and evidence begin.
>
>
> >- "Principled" reweighting is not well justified. The authors describe their reweighting as "principled," but …
>
> We thank the reviewer for this important comment. We agree that the word “principled” may be too strong for the current presentation of Eq. 11, since the exact multiplicative functional form and the exponents $\gamma_1, \gamma_2, \gamma_3$ are not uniquely derived from an optimality principle. We will revise the manuscript to avoid overstating this point and describe the reweighting more precisely as a structured, CL-motivated adaptive reweighting rule.
>
> Our intention was to design $\alpha_c$ as a compact parameterization of three continual-learning-specific imbalance factors: task progression, class recency, and replay-buffer frequency. These factors arise from the gradient-imbalance analysis in Section 3.2: as tasks progress, previous classes become increasingly underrepresented; older classes are more vulnerable to drift; and replayed samples may be repeatedly exposed due to the limited buffer. The multiplicative form in Eq. 11 is therefore not meant to be the only possible formulation, but a simple factorized correction where each term controls one source of imbalance. Equivalently, in log space, the formulation corresponds to an additive combination of these CL-specific correction factors, with $\gamma_1, \gamma_2, \gamma_3$ controlling their relative strength.
>
> We also agree that, in principle, one could optimize $\alpha_c$ directly, for example, through Bayesian optimization. However, optimizing a separate weight for every class and task would introduce many additional degrees of freedom, substantially increase validation cost, and risk overfitting to a particular task sequence or benchmark. In contrast, our formulation uses only three global hyperparameters and computes class weights dynamically from observable CL statistics, making it lightweight and scalable to longer task sequences.
>
> Following the reviewer’s suggestion, we will revise the paper in two ways. First, we will clarify that Eq. 11 is an empirically motivated, structured reweighting rule rather than a uniquely derived optimal solution. Second, we will strengthen the justification by explicitly connecting each term in $\alpha_c$ to the corresponding source of gradient imbalance and by emphasizing the sensitivity analysis, which shows that performance remains stable across a range of hyperparameter values. We believe this clarification will make the scope and motivation of the reweighting strategy more accurate.

---

> ### Author Response · Authors · 2026-05-06
>
> >- Sensitivity analysis does not cover OOD accuracies. Given that the main contribution is about OOD generalization, the sensitivity analyses of hyperparameters (Figure 2) and (Figure 3) only report i.i.d. ACC and BWT. The sensitivity …
>
> We thank the reviewer for pointing this out. We agree that OOD sensitivity is essential for supporting the paper’s main claim. The current Figures 2 and 3 demonstrate that the proposed hyperparameters are stable for i.i.d. ACC and BWT, but they do not directly assess whether OOD robustness is similarly stable. We will correct this in the revision by adding OOD sensitivity results under the same corruption-based evaluation protocol used in the main experiments. Specifically, we are currently running the corresponding OOD sensitivity analysis for the reweighting hyperparameters and the buffer-selection parameter (E). Once the results become available, we will share them in the response. We will also revise the text to clearly state that the current sensitivity analysis supports standard CL stability, while the added results assess the sensitivity of the OOD generalization claim.
>
>
> >- The entire analysis in Section 3 concerns gradient imbalance and generalization in the standard sense. There is no formal argument connecting unbalanced gradient propagation specifically to OOD performance …
>
> We thank the reviewer for this sharp and substantive critique. We agree that, in the current manuscript, the connection between the gradient-imbalance analysis in Section 3.2 and OOD performance degradation is not made sufficiently explicit. The reviewer is correct that the gradient derivation itself does not require an OOD assumption; it describes a CL-induced optimization mechanism that affects both i.i.d. and OOD performance. As we acknowledged in our response to the reviewer's first concern, this OOD connection was implicit in the current draft; the argument below is the formalization we committed to provide.
>
> To address this gap, we will add an explicit margin-based argument connecting unbalanced gradient propagation to OOD degradation. Let $f_\theta(x)$ denote the feature representation and ${w_c}$ the linear classifier weights. For a sample $(x, y)$, define the classification margin as
> $$m(x) = w_y^\top f_\theta(x) - \max_{c \neq y} w_c^\top f_\theta(x).$$
> For a label-preserving corruption $q$, as used in our OOD evaluation, suppose the induced feature shift is bounded as
> $$| f_\theta(q(x)) - f_\theta(x) | \leq L_\theta \delta,$$
> where $L_\theta$ captures the local sensitivity of the representation and $\delta$ measures the corruption magnitude. Then a sufficient condition for preserving the prediction under corruption is
> $$m(x) > \max_{c \neq y} | w_y - w_c | \cdot L_\theta \delta.$$
> This makes the OOD connection explicit: clean/i.i.d. 0-1 accuracy only requires the margin to be positive, whereas robustness to covariate shift requires the margin to be large enough to tolerate the feature displacement induced by corruption. Therefore, two models can have similar i.i.d. accuracy but very different OOD accuracy if one has more compressed margins.
>
> The gradient-polarity analysis in Section 3.2 explains why such margin compression naturally arises for old classes in rehearsal-based CL. Since current-task samples dominate training, old-class weights receive only sparse positive updates through the limited buffer, while they repeatedly appear as non-target classes for new-task samples. At the same time, the encoder is primarily updated toward the current task, producing feature drift for past classes. Together, these effects reduce the old-class margin $m(x_{\text{old}})$. Under i.i.d. evaluation, this is only observable in clean 0-1 accuracy once the margin crosses zero. Under OOD evaluation, however, even positive but small margins can fail after label-preserving corruptions, leading to a disproportionately larger drop in OOD performance, which is supported by the empirical pattern in Figure 1 and Table 1.
>
> This argument also clarifies the role of each AA-RR component in improving OOD robustness in addition to reducing forgetting. The adaptive reweighting component directly counteracts the gradient imbalance that compresses old-class margins. The augmentation pipeline encourages representation stability under label-preserving transformations, which can reduce sensitivity to corruption-induced feature shifts. The correctness-guided buffer management retains stable and representative replay samples under class- and task-balance constraints, helping preserve past-class decision regions across the task sequence.
>
> Following the reviewer's suggestion, we will revise Section 3 by adding this formal connection between gradient imbalance, margin compression, and OOD degradation.

---

> ### Author Response · Authors · 2026-05-06
>
> >- Missing memory complexity analysis. The "correctness-guided buffer management" (Section 4.3) requires tracking per-sample correctness over epochs for each task, which involves storing prediction outcomes for all training …
>
> We thank the reviewer for pointing this out. We agree that the current manuscript does not explicitly analyze the memory overhead of correctness-guided buffer management, and we will add this analysis in the revision.
>
> In the current description, the correctness score is defined as $r_i=\sum_{e=1}^{E} c_{e,i},$ where $c_{e,i}$ indicates whether sample $i$ is correctly classified at epoch $e$. A literal implementation that stores all $c_{e,i}$ values would require $O(E N_t)$ binary values for the current task, where $N_t$ is the number of samples in task $t$ and $E$ is the early-epoch tracking window.
>
> However, storing the full $E \times N_t$ correctness matrix is unnecessary. In our implementation, correctness is tracked with a single cumulative counter per current-task sample: $r_i \leftarrow r_i + \mathbf{1}[\hat{y}_i = y_i],$ for $e \leq E$. Thus, the optimized temporary memory overhead during task $t$ is only $O(N_t)$ scalar counters, not $O(E N_t)$.
>
> After the task is completed, samples are selected according to their correctness scores and inserted into the buffer in sorted order within each class/task. For future buffer reallocation, we only need to retain the top-ranked samples according to this ordering. Therefore, explicit correctness scores do not need to be stored for all retained buffer samples; the ordering itself is sufficient for pruning. Consequently, the additional memory beyond a standard replay buffer is only the temporary $O(N_t)$ counters used during current-task training.
>
> For example, on Split CIFAR-100, each task contains $N_t=5000$ samples, so the additional memory is only 5000 scalar counters during training. These counters can be stored as small integers, making the overhead negligible compared with replay-image storage, model parameters, and optimizer states. For Split Mini-ImageNet and Split Tiny-ImageNet, each task contains $N_t=10000$ samples, so the overhead remains only 10000 scalar counters during training.
>
> We will revise Section 4.3 and Algorithm 1 to clarify this implementation detail, replacing wording that may suggest storing all $c_{e,i}$ values or per-buffer correctness scores with the online counter update and ordered-buffer implementation. We will also add an explicit memory-complexity comparison against standard rehearsal training to show that correctness-guided buffer management adds negligible memory overhead.

---

> > ### Comment · Reviewer_oSrw · 2026-05-08
> >
> > I thank the authors for their clarifications. However, the revision seems not to be updated (I still see the old version).

---

### Review · Reviewer_xESW · 2026-04-24

**Summary Of Contributions:**

This work proposed the "Adaptive Augmented Reweighted Replay" method for rehearsal-based approaches in continual learning tasks in image classification problem. The method proposed to use:
1) A designed reweight parameter (Eq. 11) in the training lose,
2) Improved data augmentation method,
3) Replay buffer management (special method to update replay buffer)
to improve the performance.

Experiments on CIFAR, Mini/Tiny-ImageNet datasets demonstrates the performance of the model.

**Audience:**

Yes

**Audience Explanation:**

The continual learning task is important for machine learning community with potential applications in AGI, where the knowledge is expected to be learned without forgetting previous knowledge.

**Claims And Evidence:**

Yes

**Claims Explanation:**

The proposed model is a combination of multiple individual small methods. Although the overall approach lacks theoretical analysis, the method is intuitively valid with effectiveness justified by the experiments.

**Requested Changes:**

Mostly minor:

1. The cosine similarity in Eq. 12 should not be written as $\cos(\cdot)$ to avoid confusion with the actual cosine function.
2. The tables should have top and bottom lines.

---

> ### Author Response · Authors · 2026-05-06
>
> We thank the reviewer for these helpful comments.
>
> **1. Notation in Eq. 12.** We agree that writing cosine similarity as `cos(.)` may be confused with the trigonometric cosine function. In the revised manuscript, we will replace this notation with a clearer cosine-similarity operator, e.g., `sim(·,·)` or `CosSim(·,·)`, and explicitly define it as the cosine similarity between the normalized feature representation and normalized class weight. This change will be applied consistently wherever the notation appears.
>
> **2. Table formatting.** We agree with the reviewer. In the revised manuscript, we will update all tables to include appropriate top and bottom rules, following standard formatting conventions for readability and consistency.

---

### Decision · Action_Editor_HqTd · 2026-06-08

**Recommendation:** Reject

**Additional Comments:**

Decision rationale: The paper addresses a real problem and the experimental results are solid. Reviewers and I both recognized genuine contributions in the method and the empirical findings. However, looking at what is actually needed to bring this paper to an acceptable state, the scope goes well beyond what can reasonably be called a minor revision. The promised manuscript revision was never uploaded, meaning the paper is currently in a state that predates the author response. On top of that, the required changes include new experiments (OOD sensitivity analysis, augmentation isolation ablation), substantive rewriting of the theoretical framing in Section 3, incorporation of the margin-based argument from the rebuttal, two missing citations with related work additions, and a new discussion section addressing the pre-trained model setting. Individually some of these are small; collectively they constitute a major revision. Asking the authors to do all of this under a "minor revision" designation, with the expectation that it can be verified in a camera-ready check, is not realistic.
AE is therefore recommending rejection. This is not a judgment on the merit of the underlying idea, it is a recognition that the paper needs more work than the current review cycle allows. I encourage the authors to resubmit once the concerns are fully addressed.

If the authors choose to resubmit, please carefully address all of the following.

1. Update the revised manuscript. None of the rebuttal commitments is reflected in the current PDF.

2. Revise the "first study" claim. Replace with a precisely scoped statement consistent with the existing cited literature. Cite Guo et al. (arXiv 2023) and clarify the distinction between their setting (intra-class shift during training) and the paper's focus (test-time covariate shift evaluation).

3. Incorporate the margin-based argument from the rebuttal into Section 3. The formal connection between gradient imbalance, margin compression, and OOD vulnerability must appear in the submitted paper.

4. Add a clean ablation isolating the augmentation component. Add an "Oursλ + augmentation only" variant (with reservoir sampling, no correctness-guided buffer) so that Source 2's contribution to OOD improvement can be assessed independently.

5. Add OOD metrics to Figures 2 and 3.

6. Remove "principled" from Eq. 11 and replace with accurate language. Add the prior-adjustment interpretation discussed in the rebuttal.

7. Cite He (CVPR 2024) in Sections 3.3–3.4 and explain how αc differs from that approach.

8. Add memory complexity analysis to Section 4.3 confirming O(N_t) overhead.

9. Fix notation: cos(·) → CosSim(·,·) with explicit definition. Add top/bottom table rules throughout.

10. Add a discussion of how AA-RR applies — or does not apply — to the pre-trained model CL paradigm, with component-level analysis.

11. On experimental scale. Reviewer_U6Jy suggested larger-scale datasets or longer task sequences, and the authors committed to running them. Please include them, even briefly in the appendix, would help address the Criterion 2 concern about the narrowing relevance of small-scale benchmarks.

Below are suggestions rather than requirements, but I think they would make the paper substantially stronger.

1. On the three-source framing. Sources 1 and 2 are currently presented as established findings of the paper. The more honest framing is that they are motivating hypotheses, grounded in prior literature, which then drive the design choices. Source 3 is the one the paper actually demonstrates. If the required augmentation ablation (Required Change 4) shows a clear OOD gain from augmentation alone, then Source 2 becomes a demonstrated finding too, which would be a much stronger paper. If the gain is small or negligible, that is also worth knowing and reporting. Either way, running this experiment and reporting it honestly is the right move.

2. On the correctness-guided strategy. The strategy assumes samples that are correctly classified early in training are representative and stable. This assumption may not hold when tasks are small, labels are noisy, or early training is unreliable. The "Lowest" and "Uniform" ablation variants suggest that the strategy is not universally dominant. It would be useful to discuss conditions where correctness-guided selection might underperform and whether there are practical signals an end user could use to decide when to apply it.

3. On the OOD evaluation protocol. All corruptions are applied at a fixed severity level. Showing results across multiple severity levels, even in an appendix, would considerably strengthen the robustness claims. A method that improves at medium severity but fails at high severity tells a very different story.

**Audience:**

Yes

**Audience Explanation:**

I want to be honest that I have more reservations here than the reviewers expressed, and I think it is worth laying them out even though they do not change the decision.

On one hand, TMLR's standard for this criterion is deliberately low, "at least some individuals" is a permissive bar that does not require novelty or broad impact. The from-scratch rehearsal-on-CIFAR setting still has an active community: CSReL appeared at ICLR 2025, DELTA at CVPR 2024, PCR at CVPR 2023. These are top venues accepting work in exactly this paradigm. Results on the standard CL benchmarks retain value for positioning against that body of literature.

On the other hand, the field has meaningfully shifted. In 2024–2025, the dominant paradigm in CL has moved toward pre-trained ViT and CLIP backbones, where the backbone is frozen or lightly adapted via prompts or adapters. The reweighting and buffer strategies in AA-RR are motivated by and designed for full from-scratch training. It is not obvious at all that gradient imbalance analysis (which assumes joint encoder-classifier updates) carries over to a frozen-backbone setting. The OOD evaluation also shows its age: ImageNet-C is a 2019 benchmark, and recent work, notably LAION-C (ICML 2025) and ImageNet-D (CVPR 2024), explicitly argues that corruption-based OOD scores no longer adequately reflect real-world robustness for current models. Applying ImageNet-C corruptions to 32×32 downscaled CIFAR images is a further step removed from any realistic deployment scenario.

AE's view: Criterion 2 is satisfied for now given the active niche community, but the paper should acknowledge this limitation explicitly rather than leaving readers to wonder about it. A discussion of how AA-RR relates to the pre-trained model CL paradigm, specifically which components transfer and which do not, would meaningfully improve the paper's long-term value.

**Claims And Evidence:**

No

**Claims Explanation:**

The core experimental finding, that AA-RR improves OOD accuracy over eleven baselines across three benchmarks, is supported. Two of three reviewers agree on this.

However, AE has several concerns about the accuracy and precision of the claims built around those results. A list of suggested improvements was provided in the additional comments.

The "first study" claim is wrong as written. The paper states: "To the best of our knowledge, this is the first study to show that the performance of rehearsal-based CL methods degrades significantly under distributional shift, where the i.i.d. assumption no longer holds." This cannot stand. Verwimp et al. (ICCV 2021) and Zhang et al. (NeurIPS 2022) are both cited in the paper and both provide empirical evidence that rehearsal methods overfit their sample memories and harm generalization. The paper cannot cite these works in Section 2 and then claim to be first in Section 1. The defensible version of this claim is narrower: the paper may be among the first to conduct a systematic quantitative evaluation of rehearsal-based CL under test-time corruption-based covariate shift as a standalone OOD benchmark. The current phrasing should be replaced with something precise.

The three claimed sources of overfitting are not equally supported. Source 3 (buffer imbalance) is ok, since Table 2 directly measures it and isolates its effect. Source 1 (gradient imbalance) is supported mathematically through Equations 3–5, but there is no direct empirical measurement of gradients, and the connection to OOD degradation specifically requires the margin argument from the rebuttal to be incorporated. Source 2 (spurious correlations) is the most problematic: the paper provides no direct evidence that spurious correlations are measurably present in CL buffers, and more importantly, there is no clean ablation isolating the augmentation component. The three-source framing implies empirical validation; what the paper actually provides is uneven.

The margin-based argument from the rebuttal must appear in the paper. This is the strongest theoretical contribution to the OOD-specific motivation and it is currently only visible to the reviewers. It belongs in Section 3.

"Principled" in Eq. 11 needs to go. The authors acknowledged this; the revision should follow through.

Two relevant works are missing from the related work. He (CVPR 2024) on gradient reweighting for imbalanced CIL overlaps directly with Sections 3.3–3.4 and should be cited and differentiated. Guo et al. (arXiv 2023) on OOD forgetting in CL should be cited to properly bound the novelty claim.

OOD metrics are missing from the sensitivity analyses. Figures 2 and 3 report only i.i.d. performance for a paper whose main claim is about OOD robustness. This needs to be fixed.

**Resubmission Of Major Revision:**

The authors may consider submitting a major revision at a later time.